# DELAY OF GERMINATION1 requires PP2C phosphatases of the ABA signalling pathway to control seed dormancy

Guillaume Née[1,2], Katharina Kramer[3], Kazumi Nakabayashi[1,6], Bingjian Yuan[1], Yong Xiang[1,4], Emma Miatton[1], Iris Finkemeier [2,3] & Wim J.J. Soppe[1,5,7]

The time of seed germination is a major decision point in the life of plants determining future growth and development. This timing is controlled by seed dormancy, which prevents germination under favourable conditions. The plant hormone abscisic acid (ABA) and the protein DELAY OF GERMINATION 1 (DOG1) are essential regulators of dormancy. The function of ABA in dormancy is rather well understood, but the role of DOG1 is still unknown. Here, we describe four phosphatases that interact with DOG1 in seeds. Two of them belong to clade A of type 2C protein phosphatases: ABA-HYPERSENSITIVE GERMINATION 1 (AHG1) and AHG3. These phosphatases have redundant but essential roles in the release of seed dormancy epistatic to DOG1. We propose that the ABA and DOG1 dormancy pathways converge at clade A of type 2C protein phosphatases.

[1] Department of Plant Breeding and Genetics, Max Planck Institute for Plant Breeding Research, Cologne 50829, Germany. [2] Institute of Plant Biology and Biotechnology, University of Münster, Schlossplatz 7, Münster 48149, Germany. [3] Plant Proteomics, Max Planck Institute for Plant Breeding Research, Carl-von-Linné-Weg 10, Cologne 50829, Germany. [4] Agricultural Genomics Institute at Shenzhen, Chinese Academy of Agricultural Sciences, Shenzhen 518120, China. [5] Institute of Molecular Physiology and Biotechnology of Plants (IMBIO), University of Bonn, Bonn 53115, Germany. [6] Present address: School of Biological Sciences, Royal Holloway University of London, Egham, Surrey TW20 0EX, UK. [7] Present address: Rijk Zwaan, De Lier 2678 ZG, Netherlands. Correspondence and requests for materials should be addressed to W.J.J.S. (email: w.soppe@rijkzwaan.nl)

Accurate timing of seed germination is important for the adaptation of plants to their environment. Seeds shed from plants with a life cycle adapted to seasonal changes are usually not able to germinate directly due to dormancy, which is defined as the incapacity of an intact viable seed to complete germination under favourable conditions[1]. Seeds will only germinate after dormancy has been released. Seed dormancy impacts on agricultural production and was under negative selection during domestication. Low dormancy ensures fast and uniform germination of crop seeds, but it can also lead to an unwanted early germination on the mother plant (pre-harvest sprouting) and reduced seed quality[2]. A good understanding of dormancy control will benefit both ecological understanding and crop management. The induction and release of dormancy are regulated by developmental and environmental factors. Dormancy is induced during seed maturation. Regulators of seed maturation and environmental conditions during seed development affect the strength of seed dormancy. Dormancy is released by imbibition at low temperatures (stratification) or extended dry storage of seeds (after ripening)[3, 4].

The role of hormones in dormancy and germination has been intensively studied. Abscisic acid (ABA) regulates seed maturation and is required for the induction of dormancy. Gibberellins (GA) are needed for germination. It is in particular the balance between ABA and GA that determines germination potential[5, 6]. ABA regulates dormancy by reducing the activity of protein phosphatase 2C (PP2C) clade A proteins like ABA INSENSITIVE 1 (ABI1) and ABI2[7, 8]. As a consequence these phosphatases lose their ability to inhibit the activity of class II SNF1-related protein kinase 2 (SnRK2) by dephosphorylation[9–11]. Several of these SnRKs positively control dormancy and the triple mutant snrk2.2 snrk2.3 snrk2.6 shows a loss of seed dormancy[12].

Two major dormancy genes, DELAY OF GERMINATION 1 (DOG1) and REDUCED DORMANCY 5 (RDO5), have been identified that seem to function independent from the plant hormones. RDO5 is a member of the PP2C family of protein phosphatases that does not show phosphatase activity[13–15]. The molecular function of DOG1 has not been solved and its protein lacks domains with a known function. Both genes control natural variation in seed dormancy between Arabidopsis accessions[13, 15, 16]. Mutations in DOG1 and RDO5 completely abolish or reduce seed dormancy, respectively. DOG1 is highly conserved in the plant kingdom and homologues in various crop species have been shown to control seed dormancy[17, 18]. The amount of DOG1 protein in seeds determines the time they have to be stored to release dormancy and we have shown that the protein loses its function during this after-ripening process[19]. The regulation of DOG1 is complex and involves polyadenylation, alternative splicing and self-binding of its splicing forms[20, 21]. In addition, DOG1 transcription is influenced by environmental factors occurring during maturation and in the seed bank such as low temperature, which is associated with enhanced DOG1 transcript and protein levels[19, 22–24]. DOG1 is predominantly located in the nucleus suggesting that it might function as a transcriptional regulator[19]. DOG1 has been proposed to act by a temperature-dependent alteration of the GA metabolism, leading to weakening of the endosperm[25]. In addition, it was recently shown that DOG1 influences transcript levels of genes involved in miRNA processing causing altered accumulation of miRNAs that control developmental phase transitions in Arabidopsis and lettuce[18]. These two potential functions of DOG1 are not mutually exclusive, but none of them has been conclusively proven to be part of the primary mechanism by which DOG1 regulates dormancy. Genetic and transcriptomic analyses suggested that DOG1 is likely to function independent from ABA. However, both ABA and DOG1 have to be present to induce seed

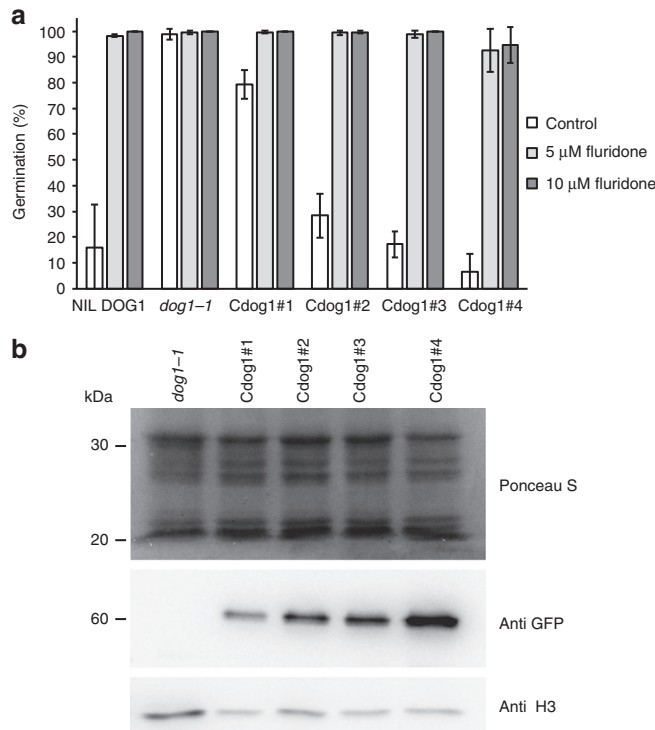

**Fig. 1** The YFP-DOG1 transgene complements the dog1-1 non-dormant phenotype. **a** Germination of freshly harvested seeds on 0.05% ethanol (control) or on 0.05% ethanol plus the indicated concentrations of fluridone. Radicle emergence was scored 7 days after sowing. Shown are averages ± s.d. of three independent batches of seeds for each genotype. **b** YFP-DOG1 protein accumulation in the complementation lines. (top) Ponceau S staining. DOG1 protein was detected using GFP antibody. After stripping, the same membrane was blotted with H3 antibody as loading control. Cdog1#1-4 represent four independent dog1-1 transformants containing the YFP-DOG1 transgene

dormancy as absence of a single one of these two regulators results in complete lack of dormancy even when the other regulator is highly accumulated[16, 19, 24, 25]. DOG1 was also shown to be required for multiple aspects of seed maturation, partially by interfering with ABA signalling components[26]. This suggests that the two pathways converge at downstream steps.

In this work, we identify four phosphatases that interact with DOG1 in dry and imbibed seeds, including the PP2C phosphatases ABA-HYPERSENSITIVE GERMINATION 1 (AHG1), AHG3 and RDO5. Genetic analysis indicates that AHG1 and AHG3 act downstream of DOG1 and are essential for its function. This suggests a model in which DOG1 controls seed dormancy by suppressing the action of specific PP2C phosphatases, which function as a convergence point of the ABA and DOG1 pathways.

## Results

**DOG1 forms complexes with protein phosphatases in seeds.** The DOG1 gene is essential for dormancy in Arabidopsis and seeds lacking functional DOG1 protein are completely non-dormant[16, 19]. The molecular mechanism by which DOG1 controls dormancy is not obvious because the primary amino acid sequence of DOG1 lacks domains with a known function. To obtain an insight into its function, we set out to identify proteins that interact with DOG1 in vivo using pull-down experiments. The antibody that we used for DOG1 detection in previous experiments[19] is not suitable for pull-down experiments.

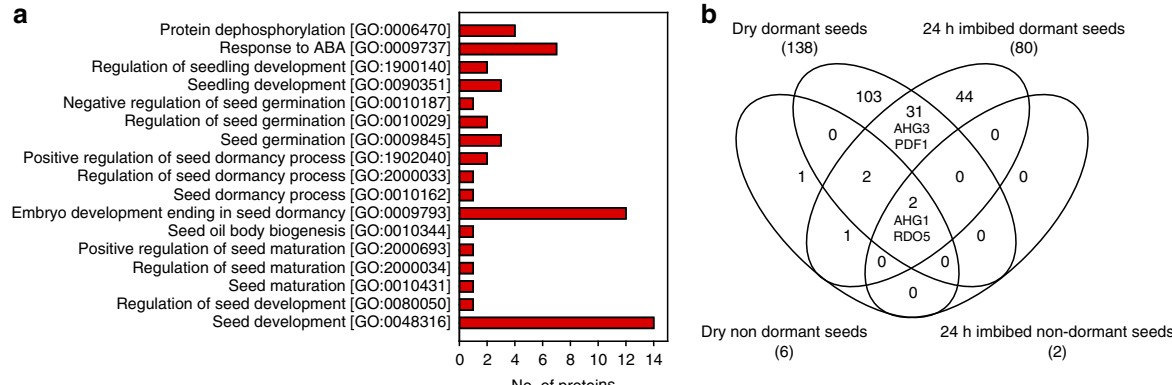

**Fig. 2** Overview of proteins that were pulled-down with DOG1 from seeds. **a** Number of YFP-DOG1 interacting proteins that are annotated with GO_biological processes related to seeds, response to ABA, and protein dephosphorylation (for clarity only GO categories with matches are shown). **b** Venn diagram showing the distribution among the different tested conditions of the 184 proteins that were identified exclusively in the YFP-DOG1 pull-down. Numbers below each condition indicate the total number of interacting proteins

Therefore, transgenic *dog1-1* plants were generated containing a yellow fluorescent protein (YFP) fused with DOG1 at the N-terminus, expressed from a strong *DOG1* promoter derived from the Cape Verde Islands (Cvi) accession (*pDOG1:YFP-DOG1*). Independent homozygous single-insertion lines were selected and analysed for their seed-dormancy level. Seeds from the *dog1-1* mutant germinated 100% directly after harvest whereas its wild-type background, NIL DOG1, only showed low germination (Fig. 1a). The transgenic lines (named Cdog1 #1, #2, #3 and #4) complemented the *dog1-1* phenotype and their dormancy levels correlated with YFP-DOG1 protein accumulation (Fig. 1). This indicated that our YFP-DOG1 fusion protein is functional. Seeds from the dormant NIL DOG1 and the complemented lines fully germinate on the ABA biosynthesis inhibitor fluridone confirming the essential role of ABA biosynthesis in dormancy even in seeds with high levels of DOG1 protein[19]. The line with the highest dormancy level and protein accumulation (Cdog1 #4) required 11 weeks of dry storage to fully release dormancy (Supplementary Fig. 1a) and was selected for pull-down experiments together with the *dog1-1* mutant as a control.

Dormant and non-dormant seed samples were taken from the same batch after different storage durations (Supplementary Fig. 1a). Proteins were isolated in their native state from dry and 24 h imbibed seeds in three biological replicates to pull-down YFP-DOG1 protein complexes using green fluorescent protein (GFP) -binding agarose beads. Gel analysis of the pull-down procedure revealed a high specificity and efficiency for YFP-DOG1 protein enrichment (Supplementary Fig. 1b). Pulled-down proteins were analysed by quantitative mass spectrometry (Supplementary Data 1), and stringent data processing (see Methods) identified 184 protein groups, which were reproducibly quantified and exclusive for the Cdog1#4 pull-downs in all three YFP-DOG1 replicates of a given condition but in none of the corresponding control pull-downs (Fig. 2). To identify proteins present in DOG1 complexes that can be relevant for its function in seed physiology we selected targets annotated with a gene ontology (GO) biological process related to seeds. We obtained 46 matches for 17 unique proteins (Fig. 2a and Supplementary Table 1) suggesting that DOG1 can interact with proteins implied in diverse seed developmental processes including dormancy. In addition, seven matches of candidates implied in ABA responses were found suggesting that DOG1 might interfere with the ABA hormone signalling pathway by complexing with ABA-related proteins (Fig. 2a and Supplementary Table 1).

Dormant seeds showed the highest number of proteins co-purifying with YFP-DOG1 (138 proteins in dry and 80 proteins in imbibed dormant seeds; Fig. 2b). Loss of seed dormancy induced a drastic decrease in the number of interacting protein (six proteins in dry and two proteins in imbibed non-dormant seeds). The decrease in number of interacting proteins upon dormancy alleviation might be a direct effect of the loss of DOG1 activity during after-ripening[19]. Two proteins, RDO5 and AHG1, were found to interact under all tested conditions (Fig. 2b). RDO5 belongs to the PP2C phosphatase family and has been described as a positive regulator of dormancy with seed-specific expression[14]. AHG1 is a PP2C clade A phosphatase[27] and members of this clade are negative regulators in the ABA signalling pathway[28, 29]. *AHG1* has a seed-specific expression pattern and shows the highest transcript levels in dry seeds among all PP2C clade A phosphatases[27] (Supplementary Fig. 2). Furthermore, among the identified interacting proteins, AHG3 and PROTEIN PHOSPHATASE 2A SUBUNIT A2 (PP2AA/PDF1) were identified in dry as well as in imbibed dormant seeds. AHG3 belongs to the same clade of PP2Cs as AHG1, while PDF1 is a scaffolding subunit A of PP2A protein phosphatase. We focussed our further studies on AHG1/3, RDO5 and PDF1 because they share relevant characteristics. They are all expressed in seeds[14, 27, 30, 31], interact with YFP-DOG1 in dormant seeds and are involved in the same molecular process: phosphoserine/threonine dephosphorylation. In addition, three of them, AHG1, AHG3 and RDO5, have been implicated in either ABA signalling or dormancy[14, 27, 30].

To confirm the interaction of DOG1 with the identified PP2C phosphatases and the PP2A phosphatase subunit, we performed a yeast two-hybrid GAL4 assay. Co-transformation of pACT2:DOG1 with pAS2:RDO5/AHG1/AHG3 restored yeast growth on selective medium and β-galactosidase activity. This restoration was also observed for pAS2:PDF1, although at a weaker level (Supplementary Fig. 3). We subsequently analysed the interaction of DOG1 with these proteins in planta using bimolecular fluorescence complementation (BiFC) in epidermis cells of *Nicotiana benthamiana* leaves. Interaction of DOG1 with AHG1, AHG3 and RDO5 was observed in the nucleus (Fig. 3). The specificity of DOG1 binding to AHG1 and AHG3 within the PP2C clade A phosphatases was analysed by testing its interaction with another phosphatase of this clade, ABI2, which had not been identified in the DOG1 pull-down assay. Restoration of YFP fluorescence was not observed in leaves co-transformed with DOG1 and ABI2 fusion proteins despite the presence of both

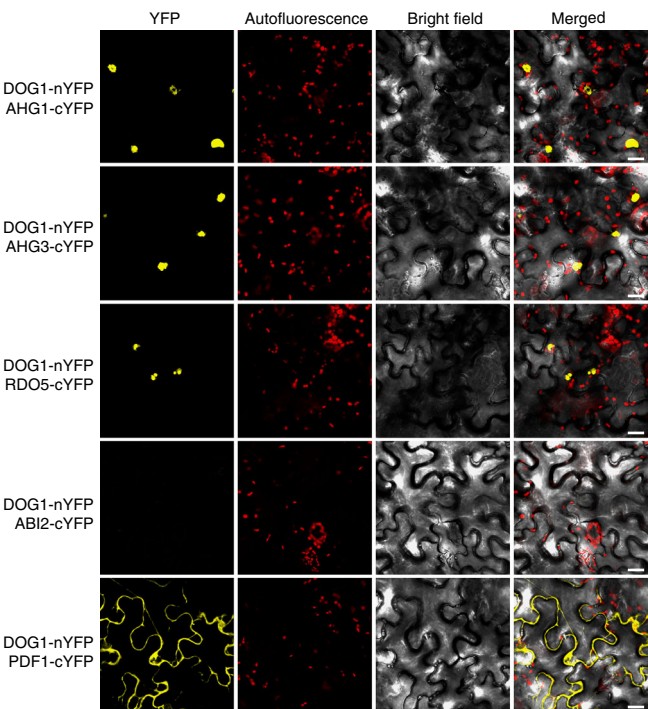

**Fig. 3** Confirmation of interactions between DOG1 and phosphatases using BiFC. BiFC assays for interactions between DOG1 and phosphatases in *Nicotiana benthamiana* leaves. Leaves were co-agro-infiltrated with constructs expressing DOG1 fused to the N- and AHG1, AHG3, RDO5, ABI2 or PDF1 fused to the C-terminus half of YFP. AHG1, AHG3 and RDO5 show strong interaction with DOG1 in the nucleus. PDF1 interacts with DOG1 in the cytoplasm. The PP2C clade A phosphatase ABI2 that was not identified in pull-down assays from seeds does not show affinity for DOG1. The reconstructed YFP fluorescence was recorded 48 h post agro-infiltration by confocal microscopy. *Red signals* indicate chloroplast autofluorescence. Scale bars, 20 μm

proteins in the transfected tobacco leaves (Fig. 3 and Supplementary Fig. 4). The BiFC assay showed interaction of DOG1 with PDF1 in the cytosol. Interaction of PDF1 and DOG1 in the cytosol might explain the weaker interaction observed in Y2H since this system requires translocation of the reconstituted complex into the nucleus to activate histidine autotrophy. Overall, these results confirmed and validated the selective interaction of DOG1 with RDO5, AHG1/3 and PDF1.

**DOG1-interacting phosphatases control seed dormancy.** DOG1 is a key seed dormancy gene and interacting proteins that are important for its function are likely to influence dormancy. Dormancy phenotypes have not been described for *pdf1* and thus we analysed its germination during seed storage. An insertion mutant in Columbia (Col) background, *pdf1*, showed enhanced seed dormancy (Fig. 4a and Supplementary Fig. 5a) compared to wild-type Col. The *pdf1* and *dog1-2* mutants were combined by crossing and selection. This double mutant completely lacked dormancy, similar to the *dog1-2* single mutant (Fig. 4a). Therefore, *dog1-2* appears to be epistatic to *pdf1*, suggesting that DOG1 functions downstream of PDF1.

Interestingly, both *ahg1* and *ahg3* mutants have been described to show enhanced sensitivity to ABA during germination and reduced germination speed[27, 30], but their dormancy loss during after-ripening has not been previously analysed. We obtained homozygous insertion mutants for *ahg1* and *ahg3* in the Col background, which were named *ahg1-5* and *ahg3-2* and which

lacked full-length transcripts of *AHG1* and *AHG3*, respectively (Supplementary Fig. 5b,c). Freshly harvested seeds of wild-type Col germinated ~70%, whereas *ahg1-5* and *ahg3-2* seeds germinated 10 and 20%, respectively (Fig. 4b). The wild-type seeds fully after-ripened after 2 weeks of storage, whereas the *ahg* single mutants needed 5–6 weeks of after-ripening to reach a germination percentage close to 100%. This demonstrated their enhanced dormancy phenotypes. Because AHG1 and AHG3 are closely related and the two highest expressed PP2C phosphatases of clade A in seeds (Supplementary Fig. 2a), we tested their redundancy by constructing the double-mutant *ahg1-5 ahg3-2*. It has been described that this double mutant shows enhanced sensitivity to ABA compared to the single mutants[27]. We observed a very strong dormancy phenotype of the *ahg1-5 ahg3-2* double mutant. Freshly harvested seeds were fully dormant and after extended storage of 22 weeks only 30% of the seeds germinated (Fig. 4b). These seeds still had the capacity to germinate because imbibition of the seeds in 100 μM $GA_{4+7}$ led to 100% germination (Fig. 4c). These results indicated that AHG1 and AHG3 have a largely redundant function in the negative regulation of seed dormancy. Since DOG1 interacts with AHG1 and AHG3 whose mutations have been shown to confer ABA hypersensitivity[27, 30], we also analysed the ABA sensitivity of *dog1* mutants. A small reduction in ABA germination sensitivity had previously been shown for the *dog1-1* mutant in the NIL DOG1 background[16]. Analysis of fully after-ripened seeds in the Col background confirmed the ABA hypersensitive germination phenotype of *ahg1-5* and *ahg3-2* single mutations and showed a decrease in ABA sensitivity for the *dog1-2* mutant (Fig. 4d).

In contrast to the other three interacting phosphatases, mutations in *RDO5* lead to strongly reduced dormancy[14] indicating an opposite role. This could be related with the lack of phosphatase activity of the RDO5 protein[13, 15].

**AHG1 and AHG3 phosphatases are required for DOG1 function.** The opposite extreme dormancy phenotypes of *dog1-2* and the *ahg1-5 ahg3-2* double mutant provide an ideal background to study their genetic relation by combining their mutations (Supplementary Fig. 6). The double mutants *dog1-2 ahg1-5* and *dog1-2 ahg3-2* were completely non-dormant. In contrast, the triple mutant *dog1-2 ahg1-5 ahg3-2* showed a very strong dormancy phenotype similar to the *ahg1-5 ahg3-2* double mutant (Fig. 4b). The triple mutant was able to germinate 100% after imbibition on 100 μM $GA_{4+7}$ (Fig. 4c). Thus, the genetic analysis demonstrated that DOG1 requires the phosphatases AHG1 and AHG3 for its function and that both phosphatases function redundantly downstream of DOG1. Seeds with high DOG1 protein levels or mutations in AHG1 or AHG3 show enhanced dormancy, suggesting that DOG1 negatively influences the action of AHG1 and AHG3. Strikingly, as for the double-mutant *ahg1 ahg3*, the triple mutant with *dog1-2* did not gradually release seed dormancy, but remained at ~10–30% germination. This suggests that AHG1 and AHG3 are essential factors in the release of dormancy by after-ripening controlled by DOG1. In addition, using stratified seeds, we could show that the triple-mutant *ahg1-5 ahg3-2 dog1-2* had a similar sensitivity to ABA as the double-mutant *ahg1-5 ahg3-2*, indicating that the *ahg1 ahg3* double mutant is epistatic to *dog1* for both dormancy and ABA sensitivity (Fig. 4d).

The phosphatase activity of most of clade A PP2Cs is inhibited in the presence of ABA by the formation of a stable complex consisting of ABA, a member of the PYR/PYL/RCAR family of ABA receptors, and PP2C. In this complex, the receptor bound to ABA hinders the active site of the phosphatase by mimicking the SnRK class II activation loop[32]. Interestingly, the seed-specific

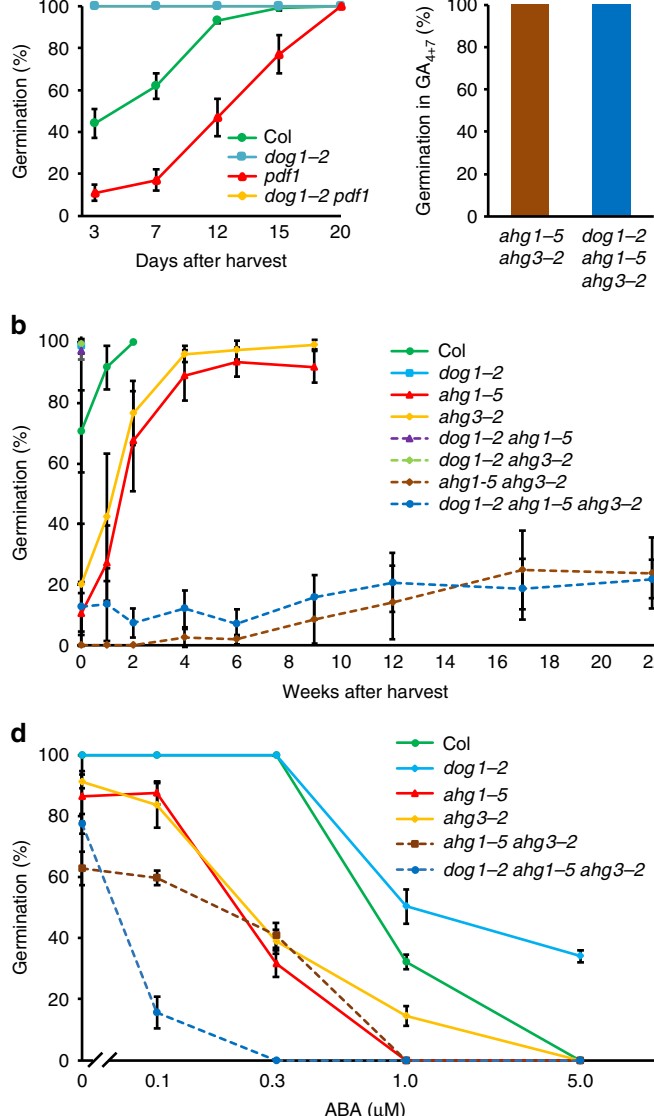

**Fig. 4** Genetic relations between mutants in *DOG1* and its interacting phosphatases. **a** Germination after different periods of dry storage of the single mutants *dog1-2* and *pdf1*, the double-mutant *dog1-2 pdf1*, and their wild-type background Col. The *dog1-2* mutant is epistatic to *pdf1* (note that the *dog1-2 pdf1* line is behind the *dog1-2* single mutant line). **b** Germination after different periods of dry storage of the single mutants *dog1-2*, *ahg1-5*, *ahg3-2*, the double mutants *dog1-2 ahg1-5*, *dog1-2 ahg3-2*, *ahg1-5 ahg3-2*, the triple mutant *dog1-2 ahg1-5 ahg3-2*, and their wild-type background Col. The *ahg1* and *ahg3* mutants have redundant roles in dormancy and are epistatic to *dog1*. The *dog1-1* mutant and the double mutants *dog1-2 ahg1-5* and *dog1-2 ahg3-2* are completely non-dormant, their symbols are overlapping in the *top left corner* of the graph (100% germination, 0 weeks after harvest). **c** Germination in 100 µM GA$_{4+7}$ of seeds from the *ahg1-5 ahg3-2* double mutant and the *dog1-2 ahg1-5 ahg3-2* triple mutant that were stored for 18 weeks. **d** Dose–response effect of ABA on germination behaviour of the same genotypes as in **b**. Seeds were after-ripened for 1 year and stratified for 3 days to enhance their germination potential. Shown are averages ± s.d. of 6–12 **a**, **b**, **c** or three **d** independent batches of seeds for each genotype

PP2CA AHG1 was shown to be resistant to inhibition by the ABA-PYR/PYL/RCAR complexes[29], which raised the question of its way of inhibition in seeds to allow downstream signalling required for dormancy. Our genetic analysis indicates that this function depends on DOG1. We therefore investigated whether

DOG1 could directly influence the catalytic activity of AHG1 or AHG3 using recombinant protein and in vitro phosphatase activity assays (Supplementary Fig. 7). Incubation of DOG1 with AHG1 or AHG3 did not lead to any significant change in their phosphatase activities even in the presence of ABA and/or the pseudophosphatase RDO5 (Supplementary Fig. 7b). It is likely that our in vitro experimental system, using purified components from *Escherichia coli*, cannot mimic the in vivo situation in seeds. A missing factor (for instance other proteins, metabolites or post-translational modifications) or a specific context (limitation in water and/or oxygen availability) putatively prevent a reliable in vitro investigation of the relationship between DOG1 and PP2Cs.

## Discussion

*DOG1* has been identified as an essential gene for seed dormancy in *Arabidopsis*. *DOG1* homologues regulating dormancy have been found in various species across the plant kingdom[17, 18, 33]. The lack of homology of DOG1 with proteins that have a known function hindered attempts to understand its molecular function. Several laboratories proposed different hypotheses for a role of DOG1 in dormancy, which were mainly based on observations at the transcript level. These included the inhibition of expression of GA-regulated genes encoding cell wall remodelling proteins and alterations in seed GA metabolism[25], reduced expression of genes associated with microRNA processing leading to reduced levels of miR156[18], and a role in seed maturation by interference with ABA signalling[26]. In this work we identified two protein phosphatases (AHG1 and AHG3) that interact with DOG1 in the seed and that are essential for its function. This provides an important direct insight into the mechanism by which DOG1 controls seed dormancy and we propose that previously identified mechanisms might function downstream of this direct mechanism.

The phosphatases AHG1, AHG3, RDO5 and PDF1 were found among the proteins that interact with DOG1 in seeds. Plants with mutations in the corresponding genes showed altered seed dormancy. Three of these phosphatases belong to the PP2C family while *PDF1* encodes one of the three scaffolding subunits of the PP2A family. Relatively little is known about these scaffolding units. One of them (*RCN1*) has a major role in the regulation of phosphatase activity and its mutant showed several defects, mainly in relation with hormone signalling[34, 35]. In contrast, mutants in the other two units (*pdf1/pp2aa2* and *pdf2/pp2aa3*) did not show obvious phenotypes[31]. Here, we identified a negative role for *PDF1* in seed dormancy. Our genetic analysis suggests that *PDF1* acts upstream of *DOG1* because the *dog1* mutant phenotype is epistatic to *pdf1*. RDO5 belongs to the PP2C phosphatase family but lacks phosphatase activity[13, 15], therefore it might act as a pseudophosphatase. In agreement with a role as pseudophosphatase, the *rdo5* mutant showed reduced dormancy in contrast to the other three identified phosphatases interacting with DOG1 that are functional and which mutants all showed enhanced dormancy.

Most importantly, DOG1 interacts in seeds with the PP2C phosphatases AHG1 and AHG3, which have been previously identified based on their ABA hypersensitive mutant phenotypes[27, 30]. In this work, we demonstrated that AHG1 and AHG3 have redundant roles in seed dormancy because the phenotype of the double mutant is much more severe than that of the single mutants (Fig. 4b). The double mutant shows extreme dormancy and did not germinate more than 20–30% after 6 months of storage. These low germination rates are not due to a loss of viability of the seeds because stored *ahg1 ahg3* double-mutant seeds could germinate 100% when imbibed in GA$_{4+7}$. Overall, this indicates a crucial role for AHG1 and AHG3 in the

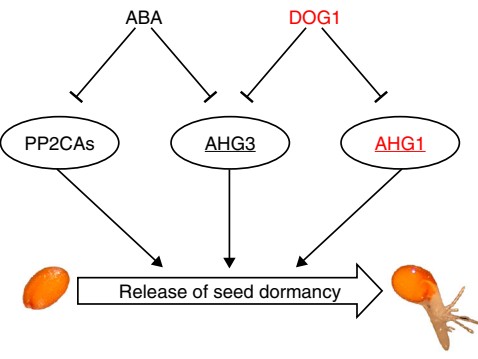

**Fig. 5** A model for the roles of DOG1 and ABA in the control of seed dormancy. DOG1 and ABA inhibit the action of distinct and overlapping members of clade A of the type 2C protein phosphatases. These PP2Cs positively regulate the release of seed dormancy and among them AHG1 and AHG3 have major roles. Seed-specific proteins are indicated in *red*. Clade A PP2C proteins are encircled. Proteins that form a complex with DOG1 in seeds are *underlined*

release of seed dormancy, while they are not required for germination per se.

We further demonstrated that these two phosphatases are essential for DOG1-dependent seed dormancy control. Our genetic data indicate that the interaction of DOG1 with AHG1 and AHG3 can negatively affect the function of these PP2Cs rather than that these phosphatases control DOG1 activity by phosphorylation. The triple mutant *dog1 ahg1 ahg3* showed the same dormancy and ABA sensitivity phenotypes as the *ahg1 ahg3* double mutant, indicating that DOG1 functions upstream of these PP2Cs. Enhanced DOG1 protein amounts[19] and absence of AHG1 and AHG3 (in the *ahg1 ahg3* double mutant) both lead to increased dormancy. Therefore, our genetic analysis suggests that DOG1 acts as a suppressor of AHG1 and AHG3 action in dormancy release.

ABA and DOG1 are both required for seed dormancy. Genetic and physiological experiments suggested that they act independently although they are likely to have common downstream targets[19]. The mechanistic basis of the crosstalk between ABA and DOG1 has been mentioned as one of the outstanding open questions in seed biology[36]. In this work, we provide data indicating that the DOG1 and ABA pathways are connected at the clade A PP2Cs.

The clade A PP2Cs function as key negative regulators of the ABA signalling pathway. In accordance with their role in ABA signalling and their expression pattern, mutants in the clade A PP2Cs show varying seed-dormancy phenotypes. Most of the nine phosphatases belonging to this clade are inhibited by the PYR/PYL/RCAR family of ABA receptors in the presence of ABA[7, 37]. Interestingly, the seed-specific clade A PP2C AHG1 was shown to be resistant to inhibition by ABA[29], although it does negatively control seed dormancy. Our observation of AHG1 and AHG3 being complexed with DOG1 in seeds and the fully epistatic behaviour of the *ahg1 ahg3* double mutant over *dog1* indicate that AHG1 and AHG3 actions are suppressed by DOG1. However, the in vitro assays did not reveal the exact molecular mechanism by which DOG1 represses phosphatase activity in seeds. It is possible that DOG1 controls PP2C action by a mechanism different than catalytic activity inhibition upon binding. The inhibition of AHG1 is likely to be controlled by DOG1 and that of AHG3 by both DOG1 and ABA (Fig. 5). Other PP2Cs are inhibited by the ABA-receptor complex and were not identified as interactors of DOG1 (Figs. 2b, 3 and Supplementary Data 1).

We propose that the ABA and DOG1 pathways converge at the level of the clade A PP2C phosphatases. The inhibition of the clade A PP2Cs by both ABA and DOG1 is supported by the reduced sensitivity to ABA of the *dog1* mutant (Fig. 4d). The functional redundancy of AHG1 and AHG3 in seed dormancy and their common inhibition by DOG1 but different inhibition by ABA could explain previous observations about the relation between DOG1 and ABA. Enhanced expression of *DOG1* in ABA biosynthesis mutants cannot induce dormancy[19]. This is probably due to the activity of PP2Cs that are not or only poorly inhibited by DOG1 and that are still able to promote germination. Similarly, the lack of dormancy of mutants with enhanced ABA levels in a *dog1* mutant background can be explained by the activity of AHG1, which is probably not inhibited by the conventional ABA pathway[29].

ABA has multiple functions in plant development and stress resistance, including seed maturation and dormancy. The main function of DOG1 is to promote seed dormancy, although a role in seed maturation has recently also been demonstrated[26]. The inhibition of germination by ABA and DOG1 through both common and separate PP2Cs would enable a flexible but robust way to shape dormancy.

The role of DOG1 in seed dormancy was shown to be conserved between several dicot and monocot species. Therefore, it is likely that the mechanism of DOG1 action that we propose for *Arabidopsis* will be conserved within the plant kingdom and could have evolved as an early adaptation of seed plants to survive seasonal conditions.

## Methods

**Plant material and growth conditions**. The *dog1-1* mutant is in the NIL DOG1 background, which is a near isogenic line that contains the *DOG1* allele from Cvi in a Landsberg *erecta* (L*er*) background[16]. The *dog1-2* mutant is in the Col background[19]. *AHG1*, *AHG3* and *PDF1* insertion lines were obtained from the NASC collection with the following seed stock numbers: *ahg1-5*, SALK_049885C; *ahg3-2*, SALK_028132; *pdf1*, SALK_037095. The gene-specific primer sequences were obtained from the SALK SIGnAL database. PCR with reverse transcription with DNA and RNA isolated from dry or imbibed seeds was performed to confirm the homozygous mutant lines (Supplementary Fig. 5 and Supplementary Table 2). Double mutants were constructed by crosses between the single mutants *dog1-2*, *ahg1-5* and *ahg3-2* and selected from the self-pollinated progeny of F2 plants using PCR to confirm their homozygosity. The triple-mutant *dog1-2 ahg1-5 ahg3-2* was obtained by crossing the double mutants *dog1-2 ahg1-5* and *dog1-2 ahg3-2* (Supplementary Fig. 6).

Seed batches were obtained from plants cultivated on soil in a growth chamber with a 16-h-light/8-h-dark cycle (22/16 °C) with the exception of the seed batches used in Fig. 4a that were obtained from plants cultivated at lower temperature (16/14 °C), conditions that enhance dormancy[4]. Freshly harvested seeds were immediately used for experiments or stored under constant conditions (21 °C, 50% humidity, in the dark) for after-ripening treatment. All comparative germination assays presented in this work use seed batches obtained at the same time from the same environment.

**Germination assays**. About 50 seeds were plated onto a filter paper moistened with demineralized water, 100 µM GA$_{4+7}$, 0.2–3 µM ABA or 5–10 µM fluridone in Petri dishes and incubated in a growth chamber (12 h light/12 h dark, 25/20 °C cycle). Stock solutions of ABA, fluridone and GA were dissolved in ethanol (final concentrations in the assays were 0.05%). When imbibed on chemicals, the control corresponds to 0.05% ethanol in water. Radicle emergence was scored after 7 days.

**Construction of YFP-DOG1 transgenic lines**. Binary constructs were prepared using the Gateway Technology (Invitrogen). A chimeric DNA fragment of *YFP* and a genomic *DOG1* fragment from Cvi (ATG to 1.1 kb downstream of the stop codon of alpha/delta splicing variants) was generated by fusion PCR using the adapter sequence 5′-ccgcagcagcccccttcacc-3′ and cloned into the pENTR/D-TOPO vector. The *DOG1* promoter region from Cvi of 2.17 kb (corresponding to the upstream region of L*er* genomic fragment[19, 21]) was inserted in the entry clone pENTR:YFP-DOG1g_Cvi using the unique *Not*I site. The binary construct pGWB1:ProDOG1:YFP-DOG1g_Cvi was produced from the above mentioned entry clone and the destination vector pGWB1[38] by LR reaction. The resulting *pDOG1_Cvi:YFP-DOG1$_{Cvi}$* construct was introduced by electroporation into *Agrobacterium tumefaciens* strain GV3101, which was subsequently used to transform *dog1-1* mutant

plants by floral dipping[39]. Independent homozygous single insertion lines were selected based on their antibiotics resistance and genotyping by PCR.

**Gene constructs**. Full-length coding DNA sequence (CDS) for AHG1, AHG3 and PDF1 were amplified from cDNA of dry Ler seeds using gene-specific primers including or not the stop codon. The PCR product was extended using the attB adapter sequences and cloned into pDONR201 or 207 (Invitrogen) using BP reactions. The pDONR207: RDO5 plasmid containing the CDS (including stop) of Ler RDO5 as well as the cloning of N-terminus truncated AHG3 (starting at amino acid position N88) was already described[14, 15]. To obtain the DOG1 expression clone for recombinant protein production, the full-length CDS of β-DOG1 from Cvi was amplified from the entry clone[21] using gene-specific primers including a BamHI site extension in the reverse primer. A synthetic double-strand DNA fragment containing an NcoI restriction site, a STREP-tag and a linker sequence was generated by denaturation and annealing (from 98 °C to room temperature) of complementary primers. DOG1 PCR product and the reconstituted tag sequence were digested with BamHI and NcoI respectively and the restricted fragment was purified from agarose gel. The chimeric STREP:DOG1 fragment was generated by blunt end ligation of digested products and cloned into pET16b (Novagen) using NcoI and BamHI sites. Sequences of all constructs were verified by Sanger sequencing.

**Yeast two-hybrid assay**. The CDS (including stop) of AHG1, AHG3, RDO5 and PDF1 were recombined from entry clones in the pAS2-gateway (GAL4 BD fusion) vectors (modified from Clontech) using LR reactions. The pACT2-gateway (GAL4 AD fusion; modified from Clontech) pACT2:DOG1 and pAS2:DOG1 containing the β-isoforms of DOG1 from Cvi were previously described[21]. Gal4-DNA-binding domain fusion proteins (prey) and GAL4 activation domain fusions protein (bait) vectors were transformed in the yeast strain PJ69-4alpha using a LiAc/SS carrier DNA/PEG method[40]. Co-transformed colonies were selected on selective medium (-LW) lacking Leu (L) and Trp (W). Interaction tests were performed on -LWH medium (-LWH) lacking L, W and His (H) with 5 mM 3-aminotriazole. Yeast was grown at 30 °C for 7 days. β-galactosidase assay was performed by pressing filter papers on the plates with yeast colonies. The filters were frozen in liquid nitrogen and subsequently incubated at 30 °C in a solution containing 0.1% X-gal.

**Bimolecular fluorescence complementation**. The full-length AHG1, AHG3, ABI2, PDF1 and RDO5 coding sequences (without stop codon) were recombined from entry clones in the pBatTL-B-sYFPc gateway vector (gift from Joachim Uhrig; MPIPZ Cologne, Germany) using LR reactions. The pBatTL-B-sYFPn containing DOG1 beta was already described[21]. Constructs were transformed into the Agrobacterium strain GV3101. Overnight cultures were diluted to an OD$_{600}$ of 0.5 in resuspension buffer (10 mM MgCl$_2$, 10 mM MES (pH 5.7) and 100 μM acetosyringone and injected into 4–6-week-old N. benthamiana leaves with a syringe. Fluorescence within the infiltrated regions was visualized after 2 days using a confocal laser scanning microscope CLSM Zeiss LSM 700. The exact same leaves used for imaging were frozen in liquid nitrogen and used to extract total protein for western blotting.

**Production and purification of recombinant protein**. Full-length CDS of AHG1, or N terminus truncated CDS of AHG3 were recombined from entry clones in the pDEST-HIS-MBP or pDEST17 gateway vector, respectively (Addgene). Expression constructs were introduced in BL21(DE3)pLysS and expression of fusion proteins was induced by 1 mM IPTG when bacterial culture reached an OD$_{600}$ of 0.6. After induction cells were grown at 23 °C overnight, harvested by centrifugation and stored at −80 °C until protein extraction. Cell pellets were resuspended in either HIS lysis buffer: 50 mM HEPES pH 7.5, 500 mM NaCl, 10% (v/v) glycerol, 5 mM DTT, 1 mM PMSF, and 25 mM imidazole or in STREP lysis buffer: 50 mM HEPES pH 7.5, 150 mM NaCl, 5 mM DTT and 1 mM PMSF. Cells were disrupted by sonication. The soluble protein fractions were separated from the insoluble fractions by centrifugation. Soluble protein fractions containing 6xHIS-MBP-AHG1, 6xHIS-N88AHG3 or STREP-DOG1 were applied on HisTrap (GE Healthcare, USA) or STREP-Tactin Superflow (IBA) 1 ml column. The purification process was performed and monitored using AKTAprime and Primeview software (GE Healthcare). Washes were performed with resuspension buffer and elution with the resuspension buffer supplemented with 500 mM imidazole or 5 mM desthiobiotin for HIS and STREP purification, respectively. Excess of salts and elution agents were removed from purified proteins by desalting against 50 mM HEPES pH 7.5, 150 mM NaCl, 10% (v/v) glycerol using PD-10 columns (GE Healthcare). Desalted purified protein were concentrated to ~2 mg ml$^{-1}$ using Amicon Ultra-15 Centrifugal Filter Unit with Ultracel-10 membrane (Millipore) and stored at −80 °C. N-truncated recombinant PP2C or MBP-PP2C fusion has previously been described to be active and suitable for PP2C inhibition assays[29, 41]. Production and purification procedures of 6xHIS-RDO5 were already described[15]. All purification steps were monitored by SDS–PAGE and all recombinant proteins could be successfully purified to full homogeneity (Supplementary Fig. 7).

**In vitro phosphatase assays**. In vitro phosphatase activity of recombinant proteins was performed with the Serine/Threonine Phosphatase Assay system (Promega, USA) using the synthetic RRA(pT)VA phosphorylated peptide as a substrate in a final volume of 50 μl in half-area, flat-bottom 96 wells plate. Assays were performed as follows: 0.5 μM of purified recombinant 6xHIS-MBP-AHG1 or 6xHIS-N88AHG3 were incubated for 20 min at room temperature in 50 mM HEPES buffer pH 7.5; 10 mM MgCl$_2$ in the absence or presence of purified recombinant STREP-DOG1 or/and purified 6 × HIS-RDO5 or/and 50 μM ABA, before starting the phosphatase activity assay by addition of the synthetic substrate at a final concentration of 50 μM. After 10 min of reaction time the activities were stopped by addition of 50 μl of molybdate dye solution. Dye was developed for 30 min, and absorbance was read at 630 nm on a multi scanspektrum (Thermo) 96 well plate reader. Specific activities were calculated using serial dilution of free phosphate as standard. In this experimental setup our recombinant purified PP2Cs, when incubated alone, displayed specific activities of 45.77 ± 4.29 and 132.35 ± 9.6 pmol of Pi min$^{-1}$μg$^{-1}$ for HIS-MBP-AHG1 and HIS-N88AHG3 respectively. Full length 6× HIS-RDO5 was already described as a PP2C without phosphatase activity[15].

**Protein extractions**. One hundred milligram of seeds (dry weight) were pulverized in liquid nitrogen with mortar and pestle. Soluble or total proteins were directly extracted from the powder in native or denaturing conditions. For native extracts, soluble proteins were resuspended for 2 h at 4 °C under constant rotation in a native buffer containing 50 mM HEPES pH 7.5; 150 mM NaCl, 1 mM EDTA; 1 mM EGTA; 0.25% (w/v) Triton X-100; 5 mM DTT; 50 U ml$^{-1}$ DNase I (Roche); 5 U ml$^{-1}$ RNase A (Machery Nagel); 1 U ml$^{-1}$ macerozyme (Boehringer-Mannheim). For total extracts, proteins were solubilized in a buffer containing 50 mM HEPES pH 7.5; 2.5% (w/v) SDS; 5 mM DTT. Both soluble and total protein extraction buffers were supplemented with 1% (v/v) of protease inhibitor cocktail special plant (Sigma) and 1% (v/v) phosphatase inhibitor cocktail 2 and 3 (Sigma). Protein fractions from seed extracts were purified by repeated centrifugation until lipid free samples were obtained. Protein concentrations of extracts were quantified by BCA protein assays (Thermo) against BSA standard dilution curve and directly used for downstream applications.

**Protein gel and immuno-detection methods**. Proteins (10 μg of total extract unless otherwise stated in the figures legends) were separated using NuPAGE® Bis-Tris gels (Thermofischer) and MES buffer. Protein gels were stained using either Oriole™ fluorescent gel stain (Biorad), or colloidal Coomassie Brilliant Blue. For Western blot, proteins were transferred on immoblion-P PVDF membrane (Milipore) using semidry transfer. After transfer proteins were stained with Ponceau S before blocking with 3% BSA. Detection of YFP-tagged DOG1 in seed extracts was performed using anti-GFP (7.1 and 13.1 mixture) monoclonal antibody produced in mouse (Roche #11814460001) at a dilution of 1:1,000. Detection of C-terminal HA-cYFP and MYC-nYFP fusion protein in tobacco leaf extracts was performed using HA tag antibody produced in rabbit (Abcam ab9110) and MYC tag antibody raised in mouse (MA1 980 Thermofischer) at a dilution of 1:7,500 and 1:1,500, respectively.

Detection of histone H3 was performed using anti-H3 antibody produced in rabbit (Abcam ab1791) at a dilution of 1:12,000. Horse radish peroxidase conjugated secondary antibodies were goat anti-mouse IgG (Sigma A3562) or goat anti-Rabbit IgG (Sigma A0545) both at a dilution of 1:30,000. Amersham ECL Prime (GE Healthcare Life Sciences) was used for detection of peroxidase activity. Images were recorded using Chemidoc XP imager (Biorad). Full scan images are available in Supplementary Fig. 9.

**YFP-DOG1 pull-down procedure**. Native seed protein extracts from 5 and 29 weeks stored Cdog1#4 and dog1-1 dry and 24 h imbibed seed were used for the pull-down assays. Pull-downs were performed from three independent biological replicates each consisting of a mix of seeds from three different plants. For each pull-down, 25 μl of agarose beads coupled to a GFP antibody (GFP-Trap_A Chromoteck gta-10) corresponding to 10 μg GFP-binding capacity was equilibrated extensively in native extraction buffer. For each replicate, an adjusted quantity of 4 mg of total protein (in 1 ml final volume) was incubated with the beads under constant rotation for 2 h at 4 °C. After incubation, beads were separated from the non-bound fraction by centrifugation and washed three times using 500 μl of native extraction buffer without Triton X-100, DNase I and RNase A. Bound proteins were eluted by incubation for 5 min with 0.1% (v/v) trifluoroacetic acid. Elutions were recovered from the beads by centrifugation and immediately neutralized by an equal volume of neutralization buffer (8 M urea; 100 mM Tris-HCl pH 8). All steps of the enrichment procedure were monitored by SDS–PAGE analysis and western blotting (Supplementary Fig. 1b).

**Mass spectrometry sample preparation and analysis**. Eluted proteins for pull-down assays were reduced, alkylated and digested in solution[42]. Cysteines were reduced by adding DTT to a final concentration of 5 mM and incubation for 30 min. Subsequently, alkylation was performed by adding chloroacetamide to a final concentration of 14 mM and incubation for 30 min. The reaction was quenched by addition of DTT. Urea concentration was adjusted to 2 M by dilution with 0.1 M Tris-HCl pH 8.0, 1 mM CaCl$_2$. Trypsin digestion (1:100 enzyme-to-

protein ratio) was performed over night at 37 °C and stopped by addition of 1% formic acid. Peptides were desalted with StageTips[43] (Empore C18, 3 M), dried and directly subjected to MS measurement.

**Mass spectrometry data processing.** Raw data were processed using MaxQuant software (version 1.5.1.2, http://www.maxquant.org/) with label-free quantification (LFQ) enabled[44]. Tandem mass spectrometry spectra were searched by the Andromeda search engine against the *Arabidopsis* TAIR10_pep_20101214 database (ftp://ftp.arabidopsis.org/home/tair/Proteins/TAIR10_protein_lists/). In addition, the protein sequences of the three putative splicing variants[21] of the YFP-DOG1 (Cvi allele) fusion proteins were added to avoid bias due to DOG1 polymorphisms between Cvi and Col. Sequences of 248 common contaminant proteins and decoy sequences were automatically added during the search. Trypsin specificity was required and a maximum of two missed cleavages was allowed. The minimal peptide length was set to seven amino acids. Carbamidomethylation of cysteine residues was set as fixed, oxidation of methionine and protein N-terminus acetylation as variable modifications. Peptide-spectrum-matches and proteins were retained at a false discovery rate of < 1%.

**Analysis of the pull-down samples.** Subsequent statistical analyses of the MaxLFQ values were performed in Perseus (version 1.5.2.6, http://www.maxquant.org/). A high reproducibility between replicates for a given genotype and condition was generally observed (Supplementary Fig. 8). Quantified proteins were only retained in the final analysis if they had valid values in at least two out of the three biological replicates. MaxLFQ values intensities were $\log_2$ transformed. Two-sample $t$-tests were performed with a $P$-value cutoff of 5%. The Perseus output of the pull-down data allowed us to quantify in total 364 protein groups (Supplementary Data 1). The 364 identified protein groups were further filtered in Microsoft Excel to select proteins that were exclusively identified in all the three replicates of the YFP-DOG1 pull-downs (absent in *dog1-1* background pull-down). This filtering highlighted 184 proteins.

**In silico analysis of identified proteins.** GO tagging of proteins identified by MS was as follows: Gene IDs for all *Arabidopsis* genes annotated with seed related biological process GOs (GO:0010162 seed dormancy process; GO:0090379 secondary cell wall biogenesis involved in seed trichome differentiation; GO:0090377 seed trichome initiation; GO:0090376 seed trichome differentiation; GO:0090351 seedling development; GO:0009793 embryo development ending in seed dormancy; GO:0048838 release of seed from dormancy; GO:0048359 mucilage metabolic process involved in seed-coat development; GO:0048354 mucilage biosynthetic process involved in seed-coat development; GO:0048317 seed morphogenesis; GO:0048316 seed development; GO:0097548 seed abscission; GO:2000693 positive regulation of seed maturation; GO:2000692 negative regulation of seed maturation; GO:1902040 positive regulation of seed dormancy process; GO:1902039 negative regulation of seed dormancy process; GO:0048700 acquisition of desiccation tolerance in seed; GO:0048623 seed germination on parent plant; GO:0009845 seed germination; GO:0010187 negative regulation of seed germination; GO:0010030 positive regulation of seed germination; GO:0010029 regulation of seed germination; GO:0010431 seed maturation; GO:0010344 seed oil body biogenesis; GO:0010231 maintenance of seed dormancy; GO:0010214 seed coat development; GO:0090378 seed trichome elongation; GO:0090380 seed trichome maturation; GO:1900140 regulation of seedling development; GO:0080113 regulation of seed growth; GO:0080112 seed growth; GO:0080050 regulation of seed development; GO:0080001 mucilage extrusion from seed coat; GO:0098755 maintenance of seed dormancy by ABA; GO:2000033 regulation of seed dormancy process; GO:1990068 seed dehydration; GO:2000034 regulation of seed maturation /37 GOs in total) or GO:009737 response to ABA, GO:006470 protein dephosphorylation, and GO:006468 protein phosphorylation were downloaded from the AmiGO2 database (http://amigo.geneontology.org/amigo). The proteins identified by pull-down were matched against these GO annotations.

**Data availability.** The mass spectrometry proteomics data have been deposited to the ProteomeXchange Consortium (http://proteomecentral.proteomexchange.org) via the PRIDE partner repository[45] with the data set identifier PXD006347. The authors declare that all other data supporting the findings of this study are available within the manuscript and its Supplementary Information files or are available from the corresponding author upon request.

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

## Acknowledgements

We thank Christina Philipp for technical assistance, Anne Harzen for mass spectrometry sample preparation and Regina Gentges for the propagation of plant material. We also thank Maarten Koornneef for critical reading of the manuscript. This work was supported by the Max Planck Society.

## Author contributions

G.N., I.F. and W.J.J.S. planned this study. G.N., K.N., B.Y., Y.X., E.M. and W.J.J.S. carried out the experimental work. K.K. performed MS measurements. G.N., K.K. and I.F. performed MS data analysis. G.N. and W.J.J.S. wrote the paper.

## Additional information

**Competing interests:** The authors declare no competing financial interests.

