## [Peer Review File · Nature Communications]

Reviewers' comments:

Reviewer #1 (Remarks to the Author):

This work by Nee et al. describes about the identification of DOG1 interacting proteins. They identified several protein phosphatases, in which ABA-related PP2Cs were included. Genetic analysis using *dog1ahg1ahg3* showed that DOG1 acts as an upstream regulator of PP2C. These results bring us new insights into the crosstalk between the regulation of seed dormancy and ABA signaling. However, there are some major problems in this manuscript as follows.

[Major points]

1. The most significant problem is that the molecular mechanism(s) between DOG1 and PP2C is still unclear. Further evidence will be required to support a proposed model in Fig. 5. In this model, DOG1 interacts with PP2Cs in seeds, however, pull-down assay and Y2H assay are not sufficient to support *in vivo* interaction between DOG1 and PP2Cs in seeds. Additional experiments will be required, such as BiFC or Co-IP experiments in seeds.
2. In addition, biochemical evidence is lacked in this manuscript. This means that arrows to AHG1/AHG3 from DOG1 are not fully supported in this work. Authors should measure PP2C activity in the presence or absence of DOG1 proteins, or in the presence or absence of ABA and DOG1. Furthermore, measurement of SnRK2 activity in WT and *dog1* will help to monitor PP2C activity *in vivo*.
3. Some other phosphatases, RDO5 and PDF1, were identified in this study. However, there are no information about the relationship between RDO5, PDF1, AHG1 or AHG3. For example, AHG1 and RDO5 competes each other in terms of DOG1 interaction, or RDO5 might inhibit AHG1-dependent dephosphorylation of target proteins?

[Minor points]

1. P.3, L56, "SNF1-related kinases (SnRKs)" should be "SNF1-related protein kinase 2 (SnRK2)".
2. P9, L195, "The loss of dormancy of *ahg1* and *ahg3* mutants during after-ripening had not been previously analysed."
However, it had been already mentioned that AHG1 or AHG3 were involved in the regulation of seed dormancy (Yoshida et al. 2006, Nishimura et al. 2007). This point should be mentioned in the manuscript.
3. P.11, L254-255, Reference(s) should be cited.
4. P.13, L302, "The inhibition of AHG1 is likely to be controlled by DOG1 and that of AHG3

by both DOG1 and ABA (Fig. 5)."

If AHG1 would be regulated by DOG1, not by ABA receptors, why ahg1-5 showed ABA-hypersensitive phenotype as shown in previous studies?

5. P.13, L305, "This is probably due to the activity of PP2Cs that are not or only poorly inhibited by DOG1 and that are still able to promote germination."

If so, it is hard to understand how DOG1 regulates seed dormancy through PP2C.

Reviewer #2 (Remarks to the Author):

The major claim of this paper is that DOG1, a plant-specific protein of unknown biochemical function, physically interacts with the specific set of PP2Cs and imposes seed dormancy.

The biochemical function of DOG1 has been a mystery and the focal point of seed dormancy research by the international seed research community. Therefore, this [redacted] manuscript addresses a very important question in seed biology. There is no question about the novelty of this report.

The discovery in this [redacted] work is directly associated with the mechanisms of ABA signal transduction and will be of great interest to many researchers in the wider field.

The originality of this paper, in addition to the discovery of DOG1 function and in comparison with that of the accompanying paper, is the depth of genetic evidence, which clearly demonstrates the critical roles and redundant function of AHG1 and AHG3 in seed dormancy downstream of DOG1.

This work has finally addressed the long-lasting question about the merging point between the ABA and DOG1 pathways of seed dormancy, which is well summarized in Figure 5 in a concise manner.

While the contents of this paper are in high quality, the following points still need to be addressed before publication:

The labeling YFP-DOG1 or YFP:DOG1 (hyphen or colon) needs consistency throughout the text and figures. Either way is fine if the later does not give an impression of a promoter construct. (pDOG1:YFP-DOG1 could be mentioned at the beginning of Results.)

Line 77: delete "hormone" from "GA hormone metabolism"

Line 126: The word "lane" can be changed to "fraction" (in other places also).

Figures should be self-explanatory. Abbreviations (e.g. "Cdog1" in Figure1) should be explained in the figure legends.

Supplementary Figure 1 is not very clear. I would suggest the authors make some efforts so that readers will be able to refer to this figure. I realize that this will be online-only publication and readers will be able to magnify images electronically. However, the authors could still make some efforts (e.g. just enlarging the original images, etc.). Otherwise, the impression of Supplementary Figure 1 could dilute the perception of Figure 1, which contains convincing results.

The symbols of the non-dominant mutants in Figure 4b are hard to find or be distinguished from each other.

Overall, and to add a subjective note, I think that the conclusions drawn from this [redacted] paper will significantly advance thinking in the field and will be recognized as a historical breakthrough in seed dormancy research.

Reviewer #3 (Remarks to the Author):

DOG1 has been extensively proven to be a key player, together with ABA, in the regulation of seed dormancy. However its precise function remained unknown and current evidence indicated that it acts on seed dormancy, independently of ABA action. The main interest of this manuscript is to provide novel information about DOG1 mechanism of action through interaction with PP2C phosphatases, which are core components of ABA signaling network. By in vivo pull-down experiments of YFP:DOG1 protein complexes, the authors identified 2 PP2C belonging to clade A, AHG1 and AHG3, which are negative regulators of ABA signaling, RDO5 another PP2C previously to act as a positive regulator of seed dormancy and PDF1, the protein phosphatase subunit A2. Protein interactions were then confirmed by Y2H. [redacted]

The work has been competently done and the manuscript is quite easy to read. The data on interacting protein identification and genetic analysis are solid, well presented and mostly support the conclusions. Nevertheless, there is a notable limitation in this manuscript, which does not clearly demonstrate that DOG1 interaction with clade A PP2Cs affects their phosphatase activity and/or phosphorylation of their downstream targets. Thus conclusions concerning biological effects of the DOG1/PP2C complex formation are mostly drawn from genetic data and thus remain somewhat speculative: for instance (p 12), authors say 'DOG1 probably acts as a suppressor of AHG1 and AHG3 activity'.

Another question that might be interesting to address is, since DOG1 interacts with AHG1 and AHG3 whose mutations have been shown to confer ABA hypersensitivity (Nishimura et al., 2007), why dog1 mutation does not lead to alterations in ABA sensitivity? Furthermore, if DOG1 and PP2C are interacting, overlapping expression patterns are expected at intracellular, as well as tissue levels. In the light of already available information in the literature, this would be worth commenting on.

Other minor comments:

- Figure 4b: germination curves are not visible for all genotypes.

- Figure 5 and discussion p13: 'The inhibition of AHG1 is likely to be controlled by DOG1 and that of AHG3 by both DOG1 and ABA.' Supporting evidence should be more clearly explained.
- Discussion p13: the following sentence also needs clarification: 'Similarly, the lack of dormancy of mutants with enhanced ABA in a dog1 mutant background can be explained by the activity of AHG1, which is not inhibited by ABA.' Reference/supporting data should be given.
- Supplemental Figure 1: concentrations of fluridone?
- Supplemental Figure 2: An orole fluorescent total staining for total protein is shown in the top?
- Abbreviations in figures: LFQ (Supplemental Figure 3), GCOS, TGT, BkG (Supplemental Figure 4)
- Abbreviation of primers are unclear: PDF1_RP is shown on the left of the gene and PDF1_LP on the right. AHG1_ERF1 & AHG3_ERF1?
- Supplementary Figure 7 needs additional information, since F2 were not all tested in each panel and clear conclusions are not given.

Reviewer #4 (Remarks to the Author):

Reviewing Nature Communications

DOG1 requires PP2C phosphatases of the ABA signaling pathway to control seed dormancy
By Nee et al

In order to unravel novel functions and to identify new potential substrates/interactors of delay of dormancy 1 (DOG1) protein the authors:

- 1) complemented the dog1-1 mutant with a YFP-DOG1 fusion under control of DOG1 native promoter derived from Cvi accession.
- 2) correlated seed dormancy to YFP-DOG1 protein levels
- 3) performed GFP pull down experiments from transgenic seeds (using dog1-1 seeds as control)
- 4) identified several PP2C phosphatases and a PP2A scaffolding subunit as DOG1 interacting proteins (using proteomics)
- 5) verified protein-protein interaction using yeast two hybrid (Y2H)
- 6) identified potential phosphorylation sites in DOG1 protein (using proteomics)
- 7) investigated dormancy of single/double/triple mutant lines (using dog1-2, pdf1, ahg1, ahg3)

major/minor points:

+ lines 134-135: "most of them will be secondary interaction partners" - was not tested...

+ line 161ff: "...are implied in ABA signaling and/or dormancy" - unclear in case of PP2A/PDF1 (so far mainly auxin-related functions described; please amend).

PDF1 has initially already been identified as DOG1 interaction partner (Miatton, 2012; please add citation as PhD thesis is publicly available for download)

+ lines 170- 180: unless authors confirm these potential phosphorylation sites by additional in vitro/in vivo experiments (where it might be a challenge to phosphorylate DOG1 in vitro/in vivo) the interpretation of this data is far too speculative, especially as the authors demonstrate the occurrence of the p- sites in only one replicate.

line 193: "this could be related with the lack of phosphatase activity" - please add citation Xiang et al., 2016.

This manuscript describes identification of delay of dormancy 1 (DOG1) - interacting proteins (PP2A and PP2C phosphatases) by proteomic approach and their genetic interaction. The authors demonstrate solid proteomic and genetics data, however, the current manuscript does not provide clear/clean evidence of a direct in vivo/in planta interaction, even if the expression pattern of identified proteins correlate with DOG1 expression (Nishimura et al., 2007; Miatton, 2012).

More in vivo data would be required to underline these interaction (protein co-localization in transgenic plants, BiFC tested by agroinfiltration and in protoplasts - with appropriate controls: western blotting, comparison with non-interacting PP2Cs)

Also the majority of identified proteins have already been assigned to function in seed dormancy.

Recent data of same research group (Xiang et al., 2016) demonstrates detailed analysis of one DOG1 interactor, RDO5 pseudophosphatase including genetic interaction between DOG1 and RDO5.

Reviewer #5 (Remarks to the Author):

The manuscript is primarily focused on defining the genetic interactions between the Arabidopsis DOG1 gene and two plant PP2C phosphatases, AHG1 and AHG3 in regulating seed dormancy. While the studies were prompted by the observations that DOG1 bound these and two other phosphatase subunits, RDO5 - an inactive PP2C and PDF1, a core PP2A

sc affording subunit in anti-DOG1 immunoprecipitates and by yeast two-hybrid assays. The specific significance of the phosphatase association is not explored. No evaluation of the phosphatase activity was undertaken and the potential substrates whose dephosphorylation is catalyzed by PP2C remain unknown. Thus the extent of novelty or the advance in knowledge derived from current studies is unclear.

Specific comments :

1. The authors state that ABA regulates seed dormancy and the signaling pathway downstream of ABA involves PP2C enzymes, which at least in part act to antagonize the actions of kinases (SnRKs) that control dormancy. Against this background, the molecular or functional consequences of DOG1 interactions with any of four phosphatase proteins remains unknown.
2. The authors report that AHG1 and AHG3 are both required for DOG1 function as loss of function of these genes negatively impacted seed germination. It is unclear to me how based on the observation that deletion of both genes which results in a very strong dormancy phenotype - stronger than the loss of either gene alone - would be interpreted as "AHG1 and AHG3 have redundant function in the negative regulation of seed dormancy". Overlapping perhaps but the stronger phenotype of the combined deletion surely also argues for unique contributions of these genes in the resulting phenotype.
3. As many phosphatases can form stable complexes with their substrates, the authors examined the covalent modification and identified two phosphorylation sites. However, the low frequency detection of these modifications - "single replicate ... in spite of the presence of phosphatase inhibitor" - was interpreted as functionally insignificant. It should be noted that the phosphatase inhibitor cocktails (Sigma 2 and 3) are unlikely to blunt PP2C activity to any degree as this class of enzymes were identified based on their resistance to most known phosphatase inhibitors. In addition, the simple inclusion of phosphatase inhibitors does not guarantee that the kinase responsible is activated. In the absence of the unknown upstream signal that activates the DOG1 kinase(s) and the lack of PP2C inhibition could readily account for the low levels of phosphopeptide identified. The alternative might be for the authors to assess the phosphorylation state of DOG1 in the AHG1/AHG3 double knockouts albeit the issue of activating the upstream kinase will likely not be fully resolved.
4. The manuscript contains a number of awkward sentences and typos that should be appropriately corrected.

Response to reviewers

Below we have listed our detailed answers to the comments of the reviewers (in blue).

Reviewer #1 (Remarks to the Author):

This work by Nee et al. describes about the identification of DOG1 interacting proteins. They identified several protein phosphatases, in which ABA-related PP2Cs were included. Genetic analysis using *dog1ahg1ahg3* showed that DOG1 acts as an upstream regulator of PP2C. These results bring us new insights into the crosstalk between the regulation of seed dormancy and ABA signaling. However, there are some major problems in this manuscript as follows.

[Major points]

1. The most significant problem is that the molecular mechanism(s) between DOG1 and PP2C is still unclear. Further evidence will be required to support a proposed model in Fig. 5. In this model, DOG1 interacts with PP2Cs in seeds, however, pull-down assay and Y2H assay are not sufficient to support *in vivo* interaction between DOG1 and PP2Cs in seeds. Additional experiments will be required, such as BiFC or Co-IP experiments in seeds.

Answer: The pull down assay was performed independently 16 times in total and the manuscript presents 12 independent replicates using fresh and after-ripened, dry and imbibed seeds. AHG1 and RDO5 were always found to interact with DOG1. We think that this is very strong evidence for the *in vivo* interaction between DOG1 and PP2Cs in seeds. Our Y2H assay confirmed this interaction. We have now also added BiFC experiments in the revised manuscript, as suggested by the reviewer, showing that DOG1 interacts with AHG1, AHG3, RDO5 and PDF1 *in vivo* (Figure 3). We also show that DOG1 has very low affinity for ABI2 in the BiFC experiment, indicating the selective interaction of DOG1 with a subset of the PP2C clade A phosphatases.

2. In addition, biochemical evidence is lacked in this manuscript. This means that arrows to AHG1/AHG3 from DOG1 are not fully supported in this work. Authors should measure PP2C activity in the presence or absence of DOG1 proteins, or in the presence or absence of ABA and DOG1. Furthermore, measurement of SnRK2 activity in WT and *dog1* will help to monitor PP2C activity *in vivo*.

Answer: We had indeed not included biochemical evidence in the first version of our manuscript. The arrows from DOG1 to AHG1/AHG3 in the model were supported by the interaction of DOG1 with these phosphatases and demonstrated by the genetic data, indicating that DOG1 requires AHG1 and AHG3 for its function.

We have performed extensive *in vitro* phosphatase assays to detect the influence of DOG1 on the activity of AHG1, but did not detect a significant change in activity. Because DOG1 interacts with AHG1 and our genetic data indicate that DOG1 negatively influences AHG1 and AHG3, we assume that the *in vitro* phosphatase assays miss a factor required for phosphatase activity inhibition that is present in the seed. We had not included these data in the first version of our manuscript. We have now added these experiments because we think they give more insights in the relation between AHG1/AHG3 and DOG1, although they are not conclusive.

3. Some other phosphatases, RDO5 and PDF1, were identified in this study. However, there are no information about the relationship between RDO5, PDF1, AHG1 or AHG3. For example, AHG1 and RDO5 competes each other in terms of DOG1 interaction, or RDO5 might inhibit AHG1-dependent dephosphorylation of target proteins?

Answer: Our mutant analyses showed that RDO5 has an opposite effect on dormancy compared to AHG1, AHG3 and PDF1. We also showed that AHG1 and AHG3 have redundant roles in dormancy release. We did not yet succeed to obtain direct *in vitro* data on the relation between DOG1, AHG1,

AHG3 and RDO5. The BiFC study that has been added in the revised version of our manuscript indicates that PDF1 seems to function independent of AHG1, AHG3 and RDO5 because it mainly interacts with DOG1 in the cytoplasm.

We are further studying the relations between these five proteins at the moment, but its full understanding will require substantial further experimentation and time.

[Minor points]

1. P.3, L56, "SNF1-related kinases (SnRKs)" should be "SNF1-related protein kinase 2 (SnRK2)".

Answer: We have repaired this mistake.

2. P9, L195, "The loss of dormancy of *ahg1* and *ahg3* mutants during after-ripening had not been previously analysed."

However, it had been already mentioned that AHG1 or AHG3 were involved in the regulation of seed dormancy (Yoshida et al. 2006, Nishimura et al. 2007). This point should be mentioned in the manuscript.

Answer: The paper by Nishimura et al. indeed mentions deeper dormancy for the *ahg1* mutant, while the Yoshida et al. paper about AHG3 does not discuss dormancy. However, proper seed dormancy experiments, in which the loss of dormancy during seed storage (after-ripening) was studied, have not been described in these papers. Both papers do present a graph showing lower germination speed (or germination growth efficiency) for both mutants. This is indicated by following germination during time of stratified seeds (Figure 1E in both papers). Germination speed is often correlated with dormancy, but it is a different trait.

We have mentioned the germination growth efficiency phenotypes of *ahg1* and *ahg3* mutants with references to Yoshida et al. 2006 and Nishimura et al. 2007 in the revised manuscript (lines 182-183).

3. P.11, L254-255, Reference(s) should be cited.

Answer: We have added two references.

4. P.13, L302, "The inhibition of AHG1 is likely to be controlled by DOG1 and that of AHG3 by both DOG1 and ABA (Fig. 5)."

If AHG1 would be regulated by DOG1, not by ABA receptors, why *ahg1-5* showed ABA-hypersensitive phenotype as shown in previous studies?

Answer: The PP2C clade A phosphatases probably share important parts of their downstream pathways. ABA functions by inhibiting the action of most of these phosphatases. There are no indications that ABA is able to inhibit AHG1, but our work (interaction of DOG1 with AHG1 and genetic data) indicates that DOG1 could inhibit the action of AHG1, thereby weakening the downstream pathways. We can see this weakened downstream pathway in the *ahg1* mutant seeds by their enhanced dormancy and reduced germination. Therefore, ABA, as an upstream inhibitor of other PP2C clade A phosphatases will have a stronger effect on the (already weakened) downstream pathway of the *ahg1* mutant, leading to enhanced ABA sensitivity. This means that our model predicts that addition of ABA to *ahg1* mutant seeds will have a stronger effect on seed germination than addition of ABA to wild-type seeds. The same effect is also seen when we have mutations in additional PP2C clade A phosphatases, as demonstrated in our manuscript by the double mutant *ahg3 ahg1*, which shows much stronger inhibition of germination than the single mutants.

5. P.13, L305, "This is probably due to the activity of PP2Cs that are not or only poorly inhibited by DOG1 and that are still able to promote germination."

If so, it is hard to understand how DOG1 regulates seed dormancy through PP2C.

Answer: The sentence quoted by the reviewer is an explanation of the preceding sentence, which says "Enhanced expression of *DOG1* in ABA biosynthesis mutants cannot induce dormancy".

In our model, we explain that ABA and DOG1 both enhance dormancy by inhibiting the activity of an overlapping set of PP2C clade A phosphatases. We speculate that some of these phosphatases are inhibited by both DOG1 and ABA and others uniquely by DOG1 or ABA. We think that this model can explain all the observed DOG1 and ABA phenotypes. If only ABA or only DOG1 is absent, some PP2Cs (that are not in the overlap) will not be inhibited anymore and lead to germination (absence of dormancy). In ABA biosynthesis mutants, ABA is absent and PP2Cs that are inhibited by ABA but not by DOG1 will remain active, independent of the DOG1 levels, and cause germination. That means that in this specific situation (ABA biosynthesis mutant) DOG1 cannot regulate dormancy. However, in wild-type plants DOG1 does regulate dormancy. We have previously shown that there is a feedback regulation between ABA and DOG1 (Nakabayashi et al. *Plant Cell* (2012) 24, 2826-2838).

Reviewer #2 (Remarks to the Author):

The major claim of this paper is that DOG1, a plant-specific protein of unknown biochemical function, physically interacts with the specific set of PP2Cs and imposes seed dormancy.

The biochemical function of DOG1 has been a mystery and the focal point of seed dormancy research by the international seed research community. Therefore, this [redacted] manuscript addresses a very important question in seed biology. There is no question about the novelty of this report.

The discovery in this [redacted] work is directly associated with the mechanisms of ABA signal transduction and will be of great interest to many researchers in the wider field.

The originality of this paper, in addition to the discovery of DOG1 function and in comparison with that of the accompanying paper, is the depth of genetic evidence, which clearly demonstrates the critical roles and redundant function of AHG1 and AHG3 in seed dormancy downstream of DOG1.

This work has finally addressed the long-lasting question about the merging point between the ABA and DOG1 pathways of seed dormancy, which is well summarized in Figure 5 in a concise manner.

While the contents of this paper are in high quality, the following points still need to be addressed before publication:

The labeling YFP-DOG1 or YFP:DOG1 (hyphen or colon) needs consistency throughout the text and figures. Either way is fine if the later does not give an impression of a promoter construct. (pDOG1:YFP-DOG1 could be mentioned at the beginning of Results.)

Answer: We have replaced all colons in the fusion proteins with hyphens (YFP-DOG1). We have added the name of the construct (*pDOG1:YFP-DOG1*) at the beginning of the results.

Line 77: delete "hormone" from "GA hormone metabolism"

Answer: We have deleted the word "hormone".

Line 126: The word "lane" can be changed to "fraction" (in other places also).

Answer: We have replaced "lane" with "fraction".

Figures should be self-explanatory. Abbreviations (e.g. "Cdog1" in Figure1) should be explained in the figure legends.

Answer: We have added an explanation of Cdog1 in the figure legend.

Supplementary Figure 1 is not very clear. I would suggest the authors make some efforts so that readers will be able to refer to this figure. I realize that this will be online-only publication and readers will be able to magnify images electronically. However, the authors could still make some efforts (e.g. just enlarging the original images, etc.). Otherwise, the impression of Supplementary Figure 1 could dilute the perception of Figure 1, which contains convincing results.

Answer: We agree with the reviewer that Suppl. Fig. 1 is not very clear and we have decided to remove this figure from the manuscript because it does not nicely illustrate the germination data shown in Fig. 1. We tried to enlarge the original images, but this required too much space.

The symbols of the non-dormant mutants in Figure 4b are hard to find or be distinguished from each other.

Answer: It is not possible to distinguish the symbols of the non-dormant mutants because they are overlapping. We have added the following sentence to the legend of Fig. 4b to clarify this: "The *dog1-1* mutant and the double mutants *dog1-2 ahg1-5* and *dog1-2 ahg3-2* are completely non-dormant, their symbols are overlapping in the top left corner of the graph (100% germination, 0 weeks after harvest)".

Overall, and to add a subjective note, I think that the conclusions drawn from this [redacted] paper will significantly advance thinking in the field and will be recognized as a historical breakthrough in seed dormancy research.

Reviewer #3 (Remarks to the Author):

DOG1 has been extensively proven to be a key player, together with ABA, in the regulation of seed dormancy. However its precise function remained unknown and current evidence indicated that it acts on seed dormancy, independently of ABA action. The main interest of this manuscript is to provide novel information about DOG1 mechanism of action through interaction with PP2C phosphatases, which are core components of ABA signaling network.

By in vivo pull-down experiments of YFP:DOG1 protein complexes, the authors identified 2 PP2C belonging to clade A, AHG1 and AHG3, which are negative regulators of ABA signaling, RDO5 another PP2C previously to act as a positive regulator of seed dormancy and PDF1, the protein phosphatase subunit A2. Protein interactions were then confirmed by Y2H. [redacted]

The work has been competently done and the manuscript is quite easy to read. The data on interacting protein identification and genetic analysis are solid, well presented and mostly support the conclusions. Nevertheless, there is a notable limitation in this manuscript, which does not clearly demonstrate that DOG1 interaction with clade APP2Cs affects their phosphatase activity and/or phosphorylation of their downstream targets. Thus conclusions concerning biological effects of the DOG1/PP2C complex formation are mostly drawn from genetic data and thus remain somewhat speculative: for instance (p 12), authors say 'DOG1 probably acts as a suppressor of AHG1 and AHG3 activity'.

Answer: This issue has also been raised by reviewers 1 and 5. We have performed *in vitro* phosphatase assays to detect the influence of DOG1 on the activity of AHG1, but could not detect a significant change in activity. Because DOG1 directly interacts with AHG1 and our genetic data indicate that DOG1 negatively influences AHG1 and AHG3, we assume that the *in vitro* phosphatase assays miss a factor required for phosphatase activity inhibition that is present in the seed. We had not included these data in the first version of our manuscript. We have now added these experiments because we

think they give more insights in the relation between AHG1/AHG3 and DOG1, although they are not conclusive.

Our genetic and pull-down data suggest that mutations in *DOG1* impact the phosphorylation status of genuine targets important for the control of dormancy downstream of the DOG1-PP2C hub. To obtain a global overview of phosphorylation we analysed the seed phosphoproteome from two accessions with different dormancy levels (NIL DOG1 and Col) as well as from *dog1* mutants in these backgrounds and the double mutant *ahg1-5 ahg3-2*. We present the data of this phosphoproteome analysis in the revised version of our manuscript and highlight an example of a potential downstream target with a known role in dormancy that shows altered phosphorylation levels and that could contribute to the observed dormancy phenotypes.

Another question that might be interesting to address is, since DOG1 interacts with AHG1 and AHG3 whose mutations have been shown to confer ABA hypersensitivity (Nishimura et al., 2007), why *dog1* mutation does not lead to alterations in ABA sensitivity?

Answer: The *dog1* mutation shows reduced ABA sensitivity, which is opposite to the ABA phenotypes of AHG1 and AHG3 and confirming the model that we show in Figure 7. This phenotype of *dog1* could already be seen in Figure 2C of the 2006 PNAS paper by Bentsink et al (Volume 103, 17042-17047) (the *dog1* mutant shows reduced sensitivity to ABA compared to its background NIL DOG1), although this was not discussed in this paper. To further confirm the ABA sensitivity of DOG1 in our paper, we have included a figure in our revised manuscript showing the ABA germination response of Col wild-type, *dog1-2*, *ahg1-5*, *ahg3-2* mutants, *ahg1-5 ahg3-2* double mutant and *dog1-2 ahg1-5 ahg3-2* triple mutant. The *dog1-2* mutant shows a reduced ABA sensitivity compared to Col wild-type. All other single, double and triple mutants show ABA hypersensitivity. The double mutant *ahg1 ahg3* and the triple *ahg1 ahg3 dog1* show the highest sensitivity to ABA. Overall, these data support our model shown in Figure 7.

Furthermore, if DOG1 and PP2C are interacting, overlapping expression patterns are expected at intracellular, as well as tissue levels. In the light of already available information in the literature, this would be worth commenting on.

Answer: We have mentioned the overlapping expression patterns of DOG1, AHG1 and AHG3 (line 152). In addition, we have included BiFC experiments in the revised version of our manuscript showing that the interaction of DOG1 with AHG1, AHG3 and RDO5 takes place in the nucleus, while the interaction of DOG1 with PDF1 can mainly be seen in the cytosol.

Other minor comments:

- Figure 4b: germination curves are not visible for all genotypes.

Answer: This issue was also raised by reviewer 2 and we explained that it is not possible to distinguish the symbols of the non-dormant mutants because they are overlapping. We have added the following sentence to the legend of Fig. 4b to clarify this: "The *dog1-1* mutant and the double mutants *dog1-2 ahg1-5* and *dog1-2 ahg3-2* are completely non-dormant, their symbols are overlapping in the top left corner of the graph (100% germination, 0 weeks after harvest)".

- Figure 5 and discussion p13: 'The inhibition of AHG1 is likely to be controlled by DOG1 and that of AHG3 by both DOG1 and ABA.' Supporting evidence should be more clearly explained.

Answer: An explanation (including supporting evidence and references) to this statement was given in the preceding paragraph. We have re-arranged this part of the discussion to have a better connection between this statement and the corresponding explanation.

- Discussion p13: the following sentence also needs clarification: 'Similarly, the lack of dormancy of mutants with enhanced ABA in a *dog1* mutant background can be explained by the activity of AHG1, which is not inhibited by ABA.' Reference/supporting data should be given.

Answer: We have included a reference, Antoni, R. *et al.* Selective inhibition of clade A phosphatases type 2C by PYR/PYL/RCAR abscisic acid receptors. *Plant Physiol.* **158**, 970–980 (2012). These data have been discussed already earlier in the discussion and we thought it was better not to repeat it at this point.

- Supplemental Figure 1: concentrations of fluridone?

Answer: Supplemental Figure 1 has been removed in the revised version of our manuscript.

- Supplemental Figure 2: An oriole fluorescent total staining for total protein is shown in the top?

Answer: Yes this is correct, Figure 2b is divided in two distinct panels annotated on the left. This sentence is part of the description of Figure 2b and not of the entire Figure 2.

- Abbreviations in figures: LFQ (Supplemental Figure 3), GCOS, TGT, BkG (Supplemental Figure 4)

Answer: We have given the full names of these abbreviations in the legends of Supplemental Figures 2, 12, 13, and 14.

- Abbreviation of primers are unclear: PDF1_RP is shown on the left of the gene and PDF1_LP on the right. AHG1_ERF1 & AHG3_ERF1?

Answer: We have not given the full names of the primers in the manuscript, but LP = Left Primer and RP = Right Primer. These primers are in relation to the direction of the insert, not to the direction of the gene. Therefore PDF1_LP is located on the right side of the gene and PDF1_RP is located on the left side.

The abbreviations of the other primers: EF=Expression Forward and ER = Expression Reverse. The names AHG1_ERF1 and AHG3_ERF1 were a typing mistake. We have replaced these names with AHG1_ER and AHG1_EF in the revised version of our manuscript.

- Supplementary Figure 7 needs additional information, since F2 were not all tested in each panel and clear conclusions are not given.

Answer: The legend of Supplementary Figure 7 in the revised version of our manuscript gives an extensive explanation of the figure including conclusions.

Reviewer #4 (Remarks to the Author):

Reviewing Nature Communications

DOG1 requires PP2C phosphatases of the ABA signaling pathway to control seed dormancy

By Nee et al

In order to unravel novel functions and to identify new potential substrates/interactors of delay of dormancy 1 (DOG1) protein the authors:

- 1) complemented the dog1-1 mutant with a YFP-DOG1 fusion under control of DOG1 native promoter derived from Cvi accession.
- 2) correlated seed dormancy to YFP-DOG1 protein levels
- 3) performed GFP pull down experiments from transgenic seeds (using dog1-1 seeds as control)
- 4) identified several PP2C phosphatases and a PP2A scaffolding subunit as DOG1 interacting proteins (using proteomics)

5) verified protein-protein interaction using yeast two hybrid (Y2H)

6) identified potential phosphorylation sites in DOG1 protein (using proteomics)

7) investigated dormancy of single/double/triple mutant lines (using *dog1-2*, *pdf1*, *ahg1*, *ahg3*)

major/minor points:

+) lines 134-135: "most of them will be secondary interaction partners" - was not tested...

Answer: We have indeed not tested all 184 identified proteins for their direct or indirect interaction with DOG1. Instead, we have selected four proteins for further study. We agree that the statement "most of them will be secondary interaction partners" is too strong as this has not been proven. We have therefore deleted this sentence.

+) line 161ff: "...are implied in ABA signaling and/or dormancy" - unclear in case of PP2AA/PDF1 (so far mainly auxin-related functions described; please amend).

Answer: The PP2A complex consists of a catalytic subunit and regulatory A and B subunits. PDF1/PP2AA2 is part of the regulatory A subunit. A role in ABA signalling has been demonstrated for several members of the PP2A complex. For instance, a specific catalytic subunit (PP2Ac-2) was shown to be a negative regulator of ABA signalling (Pernas et al (2007) *Plant Journal* 51, 763-778). In addition, another member of the regulatory A subunit family (RCN1) influences ABA sensitivity (Kwak et al (2002) *Plant Cell* 14, 2849-2861).

Because of the involvement of the PP2A complex in ABA signalling, we think it is likely that PP2AA/PDF1 could also be implied in ABA signalling. However, we agree with the reviewer that direct involvement of PP2AA2/PDF1 in ABA signalling has not been demonstrated (or studied). Therefore, we have changed this sentence to: "In addition three of them, AHG1, AHG3, and RDO5, are known to be implied in either ABA signalling or dormancy".

PDF1 has initially already been identified as DOG1 interaction partner (Miatton, 2012; please add citation as PhD thesis is publicly available for download)

Answer: This PhD thesis was written by a former member of our lab who is also a co-author on this manuscript. The thesis was made publicly available for download by the University of Cologne but it is not a peer-reviewed publication and contains premature conclusions. We prefer to limit our reference list to peer-reviewed publications and therefore we have chosen not to include this PhD thesis.

+) lines 170-180: unless authors confirm these potential phosphorylation sites by additional *in vitro/in vivo* experiments (where it might be a challenge to phosphorylate DOG1 *in vitro/in vivo*) the interpretation of this data is far too speculative, especially as the authors demonstrate the occurrence of the p-sites in only one replicate.

Answer: We present this data to demonstrate our technical abilities to identify DOG1 phosphorylation, which strengthens our conclusion that DOG1 does not show enhanced phosphorylation in the *ahg1 ahg3* mutant background (this is new information that we added in the revised version and which was generated using titanium dioxide enriched phosphorylated peptides).

The possibility of DOG1 phosphorylation *in vivo* was demonstrated with high confidence using mass spectrometry. This indicates that DOG1 can be phosphorylated *in vivo*. However it is right that the phosphorylated peptide was detected only in a single replicate. Therefore, we do not draw any conclusion from this observation and suggest in our manuscript that it occurs at very low stoichiometry.

line 193: "this could be related with the lack of phosphatase activity" - please add citation Xiang et

al., 2016.

Answer: The paper by Xiang et al. was not accepted for publication yet at the time we submitted this manuscript. We have now added this citation.

This manuscript describes identification of delay of dormancy 1 (DOG1)-interacting proteins (PP2A and PP2C phosphatases) by proteomics approach and their genetic interaction.

The authors demonstrate solid proteomics and genetics data, however, the current manuscript does not provide clear/clean evidence of a direct *in vivo*/in planta interaction, even if the expression pattern of identified proteins correlate with DOG1 expression (Nishimura et al., 2007; Miatton, 2012).

More *in vivo* data would be required to underline these interaction (protein co-localization in transgenic plants, BiFC tested by agroinfiltration and in protoplasts - with appropriate controls: western blotting, comparison with non-interacting PP2Cs)

Answer: This comment was also made by reviewer 1. As explained above, we think that our pull-down experiments give very strong evidence for the *in vivo* interaction between DOG1 and PP2A and PP2C phosphatases in seeds. This interaction was confirmed using Y2H. We have now added *in vivo* BiFC experiments in the revised manuscript, confirming *in vivo* interaction of DOG1 with AHG1, AHG3, RDO5 and PDF1 (in the nucleus and cytoplasm respectively). We also show that DOG1 has very low affinity for ABI2 in the BiFC experiment, indicating the selective interaction of DOG1 with a subset of the PP2C clade A phosphatases.

Also the majority of identified proteins have already been assigned to function in seed dormancy. Recent data of same research group (Xiang et al., 2016) demonstrates detailed analysis of one DOG1 interactor, RDO5 pseudophosphatase including genetic interaction between DOG1 and RDO5.

Answer: We think that a strong point of our paper is that several of the proteins that we have identified to interact with DOG1 have a known role in seed dormancy. This means that we have identified connections between the (previously unknown) function of DOG1 and known dormancy regulators, which is a big step forward in our understanding of the molecular regulation of seed dormancy.

We have included references to our recent paper (Xiang et al., 2016), which was not accepted yet at the time when we submitted the first version of this manuscript. Xiang et al. (2016) describes (among other things) the genetic interactions between DOG1 and RDO5, which is complementary information to the work that we describe in this manuscript.

Reviewer #5 (Remarks to the Author):

The manuscript is primarily focused on defining the genetic interactions between the Arabidopsis DOG1 gene and two plant PP2C phosphatases, AHG1 and AHG3 in regulating seed dormancy. While the studies were prompted by the observations that DOG1 bound these and two other phosphatase subunits, RDO5 - an inactive PP2C and PDF1, a core PP2A scaffolding subunit in anti-DOG1 immunoprecipitates and by yeast two-hybrid assays. The specific significance of the phosphatase association is not explored. No evaluation of the phosphatase activity was undertaken and the potential substrates whose dephosphorylation is catalyzed by PP2C remain unknown. Thus the extent of novelty or the advance in knowledge derived from current studies is unclear.

Answer: We have included experiments analysing phosphatase activity and potential substrates in our revised manuscript as we also discussed in the answers to reviewers 1 and 3. In short, we performed *in vitro* phosphatase assays to detect the influence of DOG1 on the activity of AHG1, but in our hands we could not detect a significant change in activity. Because DOG1 interacts in seeds with AHG1/3 and our genetic data indicate that DOG1 negatively influences AHG1 and AHG3, we assume that the *in vitro* phosphatase assays miss a factor required for phosphatase activity inhibition that is present in

the seed. We have added these experiments because we think they give more insights in the relation between AHG1/AHG3 and DOG1, although they are not conclusive.

Our genetic and pull-down data suggest that mutations in *DOG1* impact the phosphorylation status of genuine targets important for the control of dormancy downstream of the DOG1-PP2C hub. To obtain a global overview of phosphorylation we analysed the seed phosphoproteome from two accessions with different dormancy levels (NIL *DOG1* and Col) as well as from *dog1* mutants in these backgrounds and the double mutant *ahg1-5 ahg3-2*. [redacted]

The reviewer thinks that the extent of novelty or the advance in knowledge of our paper is unclear. In contrast, reviewer 2 states that there is no question about the novelty and thinks that our manuscript could be recognized as a historical breakthrough in seed dormancy research. We tend to agree with reviewer 2. *DOG1* has been identified as an essential gene for seed dormancy and its function has been speculated for over 10 years. Several mechanisms have been proposed. For instance, *DOG1* has been suggested to act by influencing the production of microRNAs (Huo et al. (2016) PNAS 113, E2199-E2206) or by inhibiting expression of GA-regulated genes (Graeber et al (2014) PNAS 111, E3571-E3580). Evidence for these mechanisms was correlative and indirect. In our present manuscript we identified interactors of the *DOG1* protein, which in combination with our genetic data and phosphoproteome analysis provide direct evidence for a function of *DOG1* in the regulation of AHG1 and AHG3 phosphatases. This work finally answers the mechanistic basis of the crosstalk between ABA and *DOG1* considered in a recent review (Bassel 2016) as one of the most outstanding questions in seed biology.

Specific comments :

1. The authors state that ABA regulates seed dormancy and the signaling pathway downstream of ABA involves PP2C enzymes, which at least in part act to antagonize the actions of kinases (SnRKs) that control dormancy. Against this background, the molecular or functional consequences of *DOG1* interactions with any of four phosphatase proteins remains unknown.

Answer: As explained above, we have added phosphoproteome analyses to our manuscript to clarify the consequences of *DOG1* interactions with the phosphatase proteins. [redacted]

2. The authors report that AHG1 and AHG3 are both required for *DOG1* function as loss of function of these genes negatively impacted seed germination. It is unclear to me how based on the observation that deletion of both genes which results in a very strong dormancy phenotype - stronger than the loss of either gene alone - would be interpreted as "AHG1 and AHG3 have redundant function in the negative regulation of seed dormancy". Overlapping perhaps but the stronger phenotype of the combined deletion surely also argues for unique contributions of these genes in the resulting phenotype.

Answer: "Genetic redundancy means that two or more genes are performing the same function and that inactivation of one of these genes has little or no effect on the biological phenotype" (Nowak et al. (1997) Nature 388, 167-171). In Figure 4c of our manuscript we show that a mutation in either *AHG1* or *AHG3* alone causes a slight enhancement of seed dormancy. Seeds from plants containing mutations in both *AHG1* and *AHG3* show very strong dormancy. These observations fit the definition of genetic redundancy. However, considering the weak dormancy phenotypes of the single mutants, both genes are not completely redundant. Therefore, we have replaced "have a redundant function" by "have a largely redundant function".

3. As many phosphatases can form stable complexes with their substrates, the authors examined the covalent modification and identified two phosphorylation sites. However, the low frequency detection of these modifications - "single replicate ...in spite of the presence of phosphatase inhibitor" - was interpreted as functionally insignificant. It should be noted that the phosphatase inhibitor cocktails (Sigma 2 and 3) are unlikely to blunt PP2C activity to any degree as this class of enzymes were identified based on their resistance to most known phosphatase inhibitors. In addition, the simple inclusion of phosphatase inhibitors does not guarantee that the kinase responsible is activated. In the absence of the unknown upstream signal that activates the DOG1 kinase(s) and the lack of PP2C inhibition could readily account for the low levels of phosphopeptide identified. The alternative might be for the authors to assess the phosphorylation state of DOG1 in the AHG1/AHG3 double knockouts albeit the issue of activating the upstream kinase will likely not be fully resolved.

Answer: We agree that the phosphatase inhibitor cocktails (Sigma 2 and 3) are unlikely to blunt PP2C activity. We added these inhibitors to quench phosphatase activities other than those of PP2C enzymes in our samples. PP2C activity is known to be strictly dependent on divalent cations such as magnesium. Our native extraction buffer contained a high concentration of EDTA. PP2C activity is known to be sensitive to EDTA and we expect that divalent cation chelation by EDTA prevents PP2C activities during our samples processing (J Biol Chem. 1999 Jul 16;274(29):20336-43. Kinetic analysis of human serine/threonine protein phosphatase 2C alpha Fjeld CC, Denu JM; Protein phosphorylation: a practical approach. edited by Grahame Hardie). We meant by adding the words "in spite of the presence of phosphatase inhibitor" cocktail 2 and 3 as well as EDTA. We have assessed the phosphorylation of DOG1 in the *ahg1 ahg3* double mutant as suggested by the reviewer. This experiment has been carried in a buffer containing 2.5 % SDS preventing any biochemical activities by immediate denaturation of protein. We used titanium dioxide enriched phosphorylated peptides to increase the resolution of our experiment on crude extract and could not detect phosphorylated DOG1 in this genotype as well as in wildtype seed. Overall, these experiments confirmed the low stoichiometry of DOG1 phosphorylation and that the reversion of this DOG1 modification is not a genuine function of AHG1/AHG3.

4. The manuscript contains a number of awkward sentences and typos that should be appropriately corrected.

Answer: The reviewer did not give specific examples of awkward sentences and typos, but we have rewritten our manuscript and tried to improve its style.

Reviewers' comments:

Reviewer #1 (Remarks to the Author):

Authors well answered to questions raised by the reviewer, and the manuscript was revised significantly. Authors added BiFC data in Fig. 3 which should be critical to support in vivo interaction of DOG1 and phosphatases. This version of manuscript provides clear evidence showing that DOG1 interacts with PP2Cs to regulated seed dormancy. This work will be an important milestone on the road to understand regulatory mechanisms of seed dormancy. However, there is a critical problem in this revision. Authors made an additional big change to the manuscript, so it is difficult to evaluate it. The problem is Fig. 5, in which phosphoproteomic data was presented. Although Fig. 5 suggested that some changes in protein phosphorylation should occur in *dog1-2* or *ahg1-5ahg3-2*, it is not so informative because the data was just explored by superficial analysis. If authors want to include phosphoproteome data in this manuscript, in-depth analysis will be required, e.g. overlap and differences between WT, *dog1-2* and *ahg1-5ahg3-2*, or discovery of new PP2C substrates and so on. Furthermore, as well as transcriptome analysis, phosphoproteome data should be confirmed in multiple ways, e.g. western-blot analysis using anti-phospho antibody, or in vitro phosphorylation/dephosphorylation assay using recombinant kinase or phosphatase. All MS data (raw data and spectrum) should be deposited to a public database.

This reviewer thinks such additional experiments, as mentioned above, would be too laborious to publish this work. This reviewer simply recommends authors to omit a phosphoproteome part from the manuscript. This data should be extremely interesting and valuable for your next publication. Otherwise authors can submit a new manuscript including phosphoproteome data with some additional information as required, statistical analysis and a series of validation experiments and so on.

Reviewer #4 (Remarks to the Author):

This paper from Nee et al. is a significantly improved version of the manuscript

Major points

1. BiFC experiments – Figure 3:

Although requested by reviewer (“with appropriate controls: western blotting, comparison with non-interacting PP2Cs”) authors avoid to demonstrate the protein expression levels by immunoblotting received after co-infiltration in *N. benthamiana*.

The shortcomings of BiFC usage in plants are widely known/discussed (Horstman et al., *Int J Mol Sci*, 2014) and suggestions provided (Kudla and Bocik, *The Plant Cell*, 2016), therefore demonstration of protein expression levels especially for “negative” controls (untagged YFP components, usage of mutated versions, proposed non-interactors) is crucial to demonstrate specificity of interaction.

In the current BiFC experiment (Figure 3) it is unclear if AHG1, DOG1, PDF1, RDO5, and CAM5 proteins are expressed at comparable levels.

Authors use Gateway (GW)-compatible vectors for BiFC experiments. Such vectors with included epitop tags (e.g. c-Myc, HA) suitable for detection by Western blotting have been published (Kamigaki et al., Plos One, 2016; Gehl et al., Molecular Plant, 2009) allowing quick recloning/repetition of transient transformation using agroinfiltration.

2) new data: [redacted] however this new figure is not relevant to the main findings/message of the manuscript and not enhancing its value. Moreover, the impact of this new data on the main finding is not discussed at all. I suggest to remove this data.

Response to reviewers

Below we have listed our detailed answers to the comments of the reviewers (in blue).

Reviewer #1 (Remarks to the Author):

Authors well answered to questions raised by the reviewer, and the manuscript was revised significantly. Authors added BiFC data in Fig. 3 which should be critical to support in vivo interaction of DOG1 and phosphatases. This version of manuscript provides clear evidence showing that DOG1 interacts with PP2Cs to regulated seed dormancy. This work will be an important milestone on the road to understand regulatory mechanisms of seed dormancy.

However, there is a critical problem in this revision. Authors made an additional big change to the manuscript, so it is difficult to evaluate it. The problem is Fig. 5, in which phosphoproteomic data was presented. Although Fig. 5 suggested that some changes in protein phosphorylation should occurs in *dog1-2* or *ahg1-5ahg3-2*, it is not so informative because the data was just explored by superficial analysis. If authors want to include phosphoproteome data in this manuscript, in-depth analysis will be required, e.g. overlap and differences between WT, *dog1-2* and *ahg1-5ahg3-2*, or discovery of new PP2C substrates and so on. Furthermore, as well as transcriptome analysis, phosphoproteome data should be confirmed in multiple ways, e.g. western-blot analysis using anti-phospho antibody, or in vitro phosphorylation/dephosphorylation assay using recombinant kinase or phosphatase. All MS data (raw data and spectrum) should be deposited to a public database.

This reviewer thinks such additional experiments, as mentioned above, would be too laborious to publish this work. This reviewer simply recommends authors to omit a phosphoproteome part from the manuscript. This data should be extremely interesting and valuable for your next publication. Otherwise authors can submit a new manuscript including phosphoproteome data with some additional information as required, statistical analysis and a series of validation experiments and so on.

Answer: We have removed the phosphoproteome data from the manuscript as suggested by the reviewer. We are in the process of uploading the mass spectrometry raw data to PRIDE and added a sentence about the availability of the data. The final PRIDE accession number will be added as soon as it is available. The Perseus output files are present in the Supplementary Dataset 1.

Reviewer #4 (Remarks to the Author):

This paper from Nee et al. is a significantly improved version of the manuscript

Major points

1. BiFC experiments – Figure 3:

Although requested by reviewer (“with appropriate controls: western blotting, comparison with non-interacting PP2Cs”) authors avoid to demonstrate the protein expression levels by immunoblotting received after co-infiltration in *N. benthamiana*.

The shortcomings of BiFC usage in plants are widely known/discussed (Horstman et al., *Int J Mol Sci*, 2014) and suggestions provided (Kudla and Bock, *The Plant Cell*, 2016), therefore demonstration of protein expression levels especially for “negative” controls (untagged YFP components, usage of mutated versions, proposed none-interactors) is crucial to demonstrate specificity of interaction. In the current BiFC experiment (Figure 3) it is unclear if AHG1, DOG1, PDF1, RDO5, and CAM5 proteins are expressed at comparable levels.

Authors use Gateway (GW)-compatible vectors for BiFC experiments. Such vectors with included epitop tags (e.g. c-Myc, HA) suitable for detection by Western blotting have been published (Kamigaki et al., Plos One, 2016; Gehl et al., Molecular Plant, 2009) allowing quick recloning/repetition of transient transformation using agroinfiltration.

Answer: We have repeated the BiFC experiment (Figure 3) and added immunoblotting as requested by the reviewer, demonstrating that all fusion proteins are expressed in transfected tobacco leaves (Supplementary Figure 4). The PP2C clade A proteins (including the non-interacting ABI2 control) are accumulated at comparable levels. We have also discussed our BiFC results with Jörg Kudla, the author of the 2016 *Plant Cell* article mentioned by the reviewer, who judged our BiFC interaction assays valid since it was (as mentioned in Kudla and Bock, *Plant Cell*, 2016) confirmed by two independent methods: Y2H and *in vivo* pull-down/MS from seeds (a method not relying on the irreversible reconstitution of a split protein).

2) new data: [redacted] however this new figure is not relevant to the main findings/message of the manuscript and not enhancing its value. Moreover, the impact of this new data on the main finding is not discussed at all.

I suggest to remove this data.

Answer: We have removed the description of DOG1 cofactors from the manuscript as suggested by the reviewer.

REVIEWERS' COMMENTS:

Reviewer #4 (Remarks to the Author):

The authors have fulfilled my requests successfully and provided a significantly improved version of the manuscript.